# PRODUCTIVE LLM HALLUCINATIONS: CONDITIONS, MECHANISMS, AND BENEFITS

## ABSTRACT

Hallucinations in large language models (LLMs) are typically regarded as harmful errors to be suppressed. We revisit this assumption and ask whether, and under what conditions, hallucinations can instead be beneficial. To address this question, we introduce **HIVE** (**H**allucination **I**nference and **V**erification **E**ngine), a task-agnostic framework that systematically evaluates the impact of hallucinated semantics across diverse tasks and models. By unifying generation, discrimination, and downstream evaluation, HIVE enables controlled comparative assessments of how hallucinations alter overall model performance. Extensive experiments on nine datasets and ten models show that hallucinations can yield substantial improvements up to **+17.2%** in accuracy especially in open-ended domains such as reasoning, biomedical, and vision language tasks. Stronger models consistently harness hallucinations, while weaker ones are more volatile. Mechanistic analyses show that hallucinations broaden semantic coverage, stabilize reasoning trajectories, and follow an inverted-U profile where moderate strength maximizes benefits across diverse tasks. These findings reframe hallucination from a defect to a controllable cognitive resource, suggesting opportunities for evaluating and training LLMs not merely to avoid hallucinations, but to exploit them constructively.

## 1 INTRODUCTION

*"Imagination is more important than knowledge. For knowledge is limited, whereas imagination embraces the entire world, stimulating progress, giving birth to evolution."*

— Albert Einstein (Einstein, 1931)

Large language models (LLMs) have achieved remarkable progress across a wide range of tasks (Chang et al., 2024; Lee et al., 2024a; Brown et al., 2020), marking a significant step toward human-like artificial intelligence (Li et al., 2024; Opedal et al., 2024; Chi et al., 2024). Yet a persistent limitation of LLMs is their tendency to produce *hallucinations* outputs that are factually incorrect or fabricated (Ji et al., 2023). Hallucination is defined as information inconsistent with the given input (Ji et al., 2023). Such hallucinations are typically treated as errors to be eliminated, particularly in applications requiring factual precision and trustworthiness (Wei et al., 2024; Lin et al., 2024; Gao et al., 2024). However, human cognition frequently involves speculative or counterfactual reasoning that departs from immediate factual constraints (Li et al., 2023). Human history, for example, is shaped by imaginative actions, such as planting seeds rather than consuming them (a choice that initially defies pragmatic logic, but ultimately produces transformative outcomes). This analogy motivates a central question: ① *Can certain hallucinations in LLMs, like human leaps of imagination, yield useful or even superior outcomes?* As illustrated in Fig. 1, we link this metaphor to an LLM case study and empirical evidence across tasks.

Findings from human studies in psychology often suggest that the boundary between genius and madness is vanishingly thin, as both are marked by departures from conventional logic, elevated associative thinking, and tolerance for uncertainty (Andreasen, 1987; Carson, 2011). Similarly, hallucinations in LLMs may not solely result from failure, but from the generative dynamics that support abstraction and creative inference, including overgeneralized pattern completion (Bubeck et al., 2023; Holtzman et al., 2020; Li et al., 2025), stochastic decoding (Kadavath et al., 2022; Holtzman et al., 2020; Welleck et al., 2020), or latent-space extrapolation (Wei et al., 2022; Press

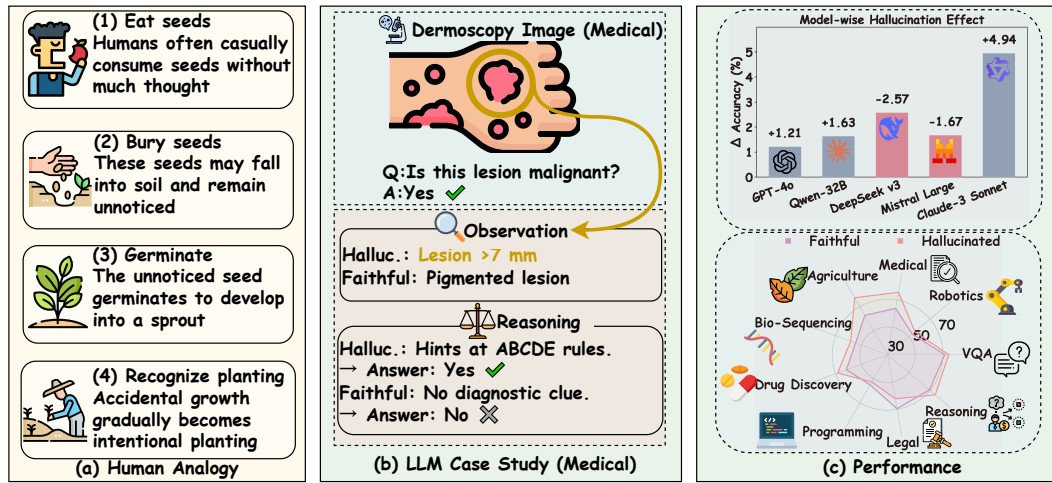

Figure 1: **From metaphor to evidence: how hallucination can help. (a) Historical analogy**: just as discarded seeds, when buried, unexpectedly gave rise to agriculture, LLM hallucinations may blossom into useful reasoning cues. **(b) Medical case study**: a hallucinated caption speculates lesion size (inaccessible), triggering the ABCDE rule and a correct answer, whereas the faithful caption provides no cue and misclassifies. **(c) Performance**: hallucination benefits vary across models (top) and tasks (bottom), showing that model capacity and task semantics shape usefulness.

et al., 2023). Viewed through this lens, hallucinations are not inherently detrimental: while some undermine reliability, others may provide valuable inductive signals, especially in open-ended or under-specified tasks (Farquhar et al., 2024; Chen et al., 2024; Park et al., 2025). This perspective gives rise two deeper questions: ② *When do beneficial hallucinations occur?* ③ *Why, in these cases, do hallucinations help rather than hurt?* Answering these questions requires a principled framework to disentangle when hallucinations act as noise and when they serve as heuristics. To answer question ②, we introduce **HIVE** (**H**allucination **I**nference and **V**erification **E**ngine), a framework designed to evaluate when and why hallucinated inputs can enhance downstream task performance. HIVE consists of three components: a caption generator, a hallucination discriminator, and a task responder. It enables controlled comparisons between raw inputs, faithful augmentations, and hallucinated augmentations. Across 9 tasks and 9 models, our experiments show that hallucinations are especially effective in perception-driven tasks like natural language inference or protein prediction, but may harm performance in rule-driven tasks like legal reasoning or code generation. Effectiveness varies by model family and task type, with some combinations yielding double-digit gains and others negligible or negative effects. Furthermore, we identify the optimal configuration: moderate temperature and token budget yield the best outcomes.

To answer question ③, our study provides two empirical observations. (I) hallucinated captions broaden semantic coverage by introducing additional, often speculative, concepts. Embedding analyses reveal that their representations spread more widely and form longer semantic tails than faithful captions, exposing models to a richer hypothesis space. (II) hallucinated captions increase semantic entropy, and correct predictions under hallucination correlate with higher-entropy representations. Hallucinations are not merely noise but serve as cues that diversify reasoning trajectories.

Our main contributions are as follows:

- **General Hallucination Framework.** We introduce HIVE, a general-purpose hallucination inference and verification framework applicable to both text-only and multimodal tasks. It unifies generation, discrimination, and downstream task-response under flexible, controllable hallucination configurations, enabling rigorous apples-to-apples comparisons across models and settings.
- **Broad Empirical Evidence.** Across 9 diverse tasks and 9 representative models, hallucination-augmented inputs deliver consistent and measurable benefits in perception-driven settings, with effect sizes shaped jointly by task semantics and model family and reaching up to +17.22%.
- **Mechanistic Insights.** Through systematic analysis at the input, process, and output levels, we demonstrate that hallucinations reshape semantic inputs, modulate inference dynamics, and

correlate semantic diversity with correct outcomes, while preserving both intra- and inter-chain convergence. These findings reframe hallucination from a defect into a controllable resource.

## 2 RELATED WORK

**Hallucination as Harm: Detection and Mitigation.** Hallucination denotes content inconsistent with the given input. Hallucination is prevalent in LLM outputs, appearing in tasks such as summarization (Zhao et al., 2020) and open-domain QA (Sadat et al., 2023). Its presence undermines reliability and safety in high-stakes domains, including healthcare (Lehman et al., 2021; Nori et al., 2023) and legal decision support (Guha et al., 2023; Bendahman et al., 2025). Two main threads dominate: detection and mitigation. For detection, works span early factuality measures and benchmarks FactCC (Kryściński et al., 2020), QAGS (Wang et al., 2020), TruthfulQA (Lin et al., 2022), Q2 (Honovich et al., 2021) to agreement and internals based methods (Manakul et al., 2023; Du et al., 2024; Su et al., 2024). For mitigation, strategies include (I) prompt/instruction tuning (Zhang et al., 2024; Liu et al., 2024; Yu et al., 2024), (II) constrained decoding (Lee et al., 2024b; Su et al., 2024; Choi et al., 2023; Mudgal et al., 2024), and (III) training or retrieval augmentation (Sennrich et al., 2024; Manevich & Tsarfaty, 2024; Lewis et al., 2020). These lines assume that hallucination is inherently harmful by nature and primarily aim to strictly suppress it.

**Hallucination as Heuristic: Emerging Evidence.** A complementary view posits that semantically relevant hallucinations can aid abstraction, creativity, or heuristic reasoning. Empirical signs appear across domains: (I) improving code vulnerability detection (Luo et al., 2025), (II) stimulating drug discovery (Yuan & Färber, 2025), and (III) enhancing multimodal representation learning via hallucination-driven contrastive signals (Jiang et al., 2024a); other works frame hallucination as a creativity mechanism or realizable propositions in context (Mizrahi et al., 2025; Jiang et al., 2024b; Chen & Wang, 2025). However, these efforts remain fragmented, highly task-specific, and generally lack systematic control over model capacity, task semantics, and the nature of injected content.

**Different from existing methods**, our method does not uncritically assume hallucinations to be inherently harmful, nor does it merely claim their benefits in a single domain (Thorne et al., 2018). In contrast, we introduce a unified evaluation paradigm that measures hallucination-induced gains across tasks, domains, and modalities, uncovering that they are especially effective in perception-driven settings but often detrimental in rule-driven ones, and further provide mechanistic insights into why hallucinations can enhance reasoning.

## 3 METHOD

### 3.1 PROBLEM FORMULATION

Our central question is how faithful versus hallucinatory semantics influence downstream model responses. To study this effect, we design our method around three principles:

① **Fair comparison.** We must ensure that the only difference between faithful and hallucinatory captions lies in the presence of hallucination itself. Thus, captions are generated from a unified source with identical prompts, temperature, and token budget. This design rules out confounds and allows us to focus on the difference in downstream performance across tasks and models, measured as the accuracy gap $\Delta(H - F)$ between hallucinated and faithful augmentations.

② **Task-agnostic hallucination generation.** Hallucinations naturally arise when an LLM is asked to produce a free-form caption of any input, regardless of modality. Some captions remain faithful, while others introduce unverifiable or speculative elements. This inherent property enables us to adapt the paradigm seamlessly across diverse textual and multimodal tasks.

③ **Reliable discrimination.** Hallucination detection is inherently imperfect even humans may disagree. To enhance robustness, we adopt an ensemble of detectors that independently judge each caption, and apply majority voting to obtain the final label. This ensemble strategy ensures that the framework remains reliable under noisy classifiers. We validate accuracy on a hallucination benchmark and a carefully curated human annotations benchmark. (See Appendix §S11).

Formally, let $x \in \mathcal{X}$ denote an input with label $y \in \mathcal{Y}$, and $f : \mathcal{X} \times \mathcal{C} \to \mathcal{Y}$ a task-specific model. Each input can be paired with a faithful caption $C_F$ or a hallucinatory caption $C_H$. Such a triplet design ensures that hallucination is the only varying factor across conditions, thus enabling a

---

**Algorithm 1:** HIVE pipeline from semantic contrastive view: Faithful (F) vs. Hallucinated (H).

---

**Input:** Dataset $\mathcal{D}$, task model $\boldsymbol{f}$, generator $G$, discriminator $D$
**Output:** Performance difference $\boldsymbol{\Delta}(H{-}F)$
**foreach** $(x, y) \in \mathcal{D}$ **do**
    Generate candidate captions $C(x) = \{c_1, \ldots, c_N\}$ via $G$ // Expose semantic diversity
    Classify each $c_i$ as Faithful or Hallucinatory using $D$ // Partition as F/H
    **if** *a contrasted pair* $\langle \boldsymbol{C}_F, \boldsymbol{C}_H \rangle$ *exists* **then**
        $y_{\text{RAW}} \leftarrow \boldsymbol{f}(x)$ // Setting for raw input
        $y_F \leftarrow \boldsymbol{f}(x \parallel \boldsymbol{C}_F)$ // Setting for faithful caption
        $y_H \leftarrow \boldsymbol{f}(x \parallel \boldsymbol{C}_H)$ // Setting for hallucinatory caption
        Record $\mathcal{L}(y_H, y)$ and $\mathcal{L}(y_F, y)$;

Compute $\boldsymbol{\Delta}(H{-}F) = \mathbb{E}_{(x,y)\sim\mathcal{D}}\big[\,\mathcal{L}(y_H, y) - \mathcal{L}(y_F, y)\,\big]$ // Hallucination effect measure

---

controlled and interpretable evaluation. Formally, we define these conditions:

$$y_{\text{RAW}} = \underbrace{\boldsymbol{f}(x)}_{\text{Raw}}, \quad y_F = \underbrace{\boldsymbol{f}(x \parallel \boldsymbol{C}_F)}_{\text{+ Faithful}}, \quad y_H = \underbrace{\boldsymbol{f}(x \parallel \boldsymbol{C}_H)}_{\text{+ Hallucinatory}}, \tag{1}$$

where $\parallel$ denotes concatenation with the task instruction. Given an evaluation metric $\mathcal{L}(\hat{y}, y)$ (instantiated as accuracy in our experiments), we quantify the hallucination effect as

$$\boldsymbol{\Delta}(H{-}F) \;=\; \mathbb{E}_{(x,y)\sim\mathcal{D}}\Big[\,\mathcal{L}\big(\boldsymbol{f}(x \parallel \boldsymbol{C}_H),\, y\big) - \mathcal{L}\big(\boldsymbol{f}(x \parallel \boldsymbol{C}_F),\, y\big)\,\Big]. \tag{2}$$

This paired comparison isolates hallucination as a controlled experimental variable and enables apples-to-apples measurement across models, tasks, and modalities.

## 3.2 WORKFLOW OF HIVE

As described in Algorithm 1, the HIVE framework consists of three modules: (I) **Caption Generator**, (II) **Caption Discriminator**, and (III) **Task Solver**. Given an input instance, the workflow proceeds as follows to ensure consistent and controlled evaluation across tasks.

The Caption Generator first takes the raw input (text, image, or structured record) and produces a set of candidate captions under a unified, task-agnostic prompt, which may include both faithful and hallucinatory semantics. The Caption Discriminator then evaluates these candidates and classifies each as either faithful ($\boldsymbol{C}_F$) or hallucinatory ($\boldsymbol{C}_H$); only contrasted pairs $\langle \boldsymbol{C}_F, \boldsymbol{C}_H \rangle$ with majority agreement among detectors are retained. Finally, the Task Solver integrates the original input $x$, one of the paired captions, and a task-specific instruction to produce the final prediction $y$. This design yields three controlled conditions Raw ($y_{\text{RAW}} = \boldsymbol{f}(x)$), +Faithful ($y_F = \boldsymbol{f}(x \parallel \boldsymbol{C}_F)$), and +Hallucinatory ($y_H = \boldsymbol{f}(x \parallel \boldsymbol{C}_H)$), allowing fair comparison.

**Caption Generator.** The Caption Generator aims to produce diverse semantic candidates that may include both faithful and hallucinatory variants. All captions are generated from a unified source using the same prompt, temperature, and token budget, ensuring that decoding hyper-parameters cannot confound attribution. Given an input $x$, the generator outputs $N$ natural-language captions describing it. Due to the inherent stochasticity of LLMs, some captions remain faithful while others introduce speculative content, which later enables the construction of contrasted F/H pairs by the discriminator. This design guarantees that both F and H captions originate from an identical generation process, providing a controlled entry point for subsequent evaluation. This generation step ensures that faithful and hallucinatory captions can later be contrasted under controlled conditions.

**Caption Discriminator.** Given the candidate captions, the Caption Discriminator determines whether each caption is faithful ($\boldsymbol{C}_F$) or hallucinatory ($\boldsymbol{C}_H$). Since hallucination detection is inherently noisy, we adopt an ensemble of multiple detectors, each providing an independent judgment. Final labels are decided via majority voting, which significantly improves robustness under noisy or imperfect classifiers. We further validate the effectiveness of the discriminator with a flipping control experiment; see Appendix §S4. Detailed implementation specifics of the individual detectors and ensemble configuration are provided in Appendix §S8 for full clarity and reproducibility.

Table 1: **Faithful (F) vs. Hallucinated (H) path accuracy.** Denote $\Delta$(H–F) as the relative accuracy performance gain $\uparrow$ or drop $\downarrow$ from the hallucinated path over the faithful path.

| Dataset | P. | GPT-4o | GPT-3.5 | Claude-3 Sonnet | DeepSeek v3 | Mistral Large | O3 | DeepSeek R1 |
|---|---|---|---|---|---|---|---|---|
| AntiCP2 | F | $54.59_{(-)}$ | $43.64_{(-)}$ | $46.88_{(-)}$ | $52.19_{(-)}$ | $70.86_{(-)}$ | $53.95_{(-)}$ | $49.87_{(-)}$ |
|  | H | $58.35_{\uparrow 3.76}$ | $48.21_{\uparrow 1.33}$ | $51.40_{\downarrow -0.43}$ | $45.01_{\downarrow -7.18}$ | $68.62_{\downarrow -2.24}$ | $50.17_{\downarrow -3.78}$ | $46.42_{\downarrow -3.45}$ |
| BBBP | F | $61.67_{(-)}$ | $60.75_{(-)}$ | $64.07_{(-)}$ | $61.60_{(-)}$ | $70.86_{(-)}$ | $73.27_{(-)}$ | $70.53_{(-)}$ |
|  | H | $68.38_{\uparrow 6.67}$ | $57.53_{\downarrow -3.22}$ | $64.05_{\uparrow 0.88}$ | $59.05_{\downarrow -2.55}$ | $68.62_{\downarrow -2.24}$ | $59.47_{\downarrow -13.80}$ | $58.88_{\downarrow -11.65}$ |
| CodeXGLUE | F | $55.15_{(-)}$ | $58.90_{(-)}$ | $49.25_{(-)}$ | $59.64_{(-)}$ | $68.17_{(-)}$ | $45.10_{(-)}$ | $51.54_{(-)}$ |
|  | H | $52.75_{\downarrow -2.40}$ | $57.40_{\downarrow -1.49}$ | $53.40_{\uparrow 4.15}$ | $58.06_{\downarrow -1.58}$ | $68.76_{\uparrow 0.59}$ | $46.06_{\uparrow 0.96}$ | $48.95_{\downarrow -2.59}$ |
| SARA | F | $62.93_{(-)}$ | $52.28_{(-)}$ | $62.07_{(-)}$ | $60.32_{(-)}$ | $70.86_{(-)}$ | $65.52_{(-)}$ | $58.62_{(-)}$ |
|  | H | $62.24_{\downarrow -0.69}$ | $54.83_{\uparrow 2.55}$ | $63.97_{\uparrow 1.9}$ | $58.60_{\downarrow -1.72}$ | $68.62_{\downarrow -2.24}$ | $62.07_{\downarrow -3.45}$ | $54.48_{\downarrow -4.14}$ |
| ProofWriter | F | $69.49_{(-)}$ | $66.55_{(-)}$ | $76.03_{(-)}$ | $73.01_{(-)}$ | $70.86_{(-)}$ | $97.76_{(-)}$ | $89.31_{(-)}$ |
|  | H | $75.00_{\uparrow 5.51}$ | $66.21_{\downarrow -0.34}$ | $76.73_{\uparrow 0.70}$ | $73.17_{\uparrow 0.16}$ | $68.62_{\downarrow -2.24}$ | $98.45_{\uparrow 0.69}$ | $92.93_{\uparrow 3.62}$ |

**(III) Task Solver.** The solver is not a novel model but a controlled interface to isolate the effect of captions on downstream predictions. We measure the isolated impact of caption faithfulness by contrasting predictions under three conditions $C \in \{\text{RAW}, C_F, C_H\}$. We use a unified prompt builder that concatenates three parts:

$$\Phi(\mathcal{I}_{\text{task}}, x, C) = \underbrace{\text{INSTRUCTION } \mathcal{I}_{\text{task}}}_{\text{Fixed}} \parallel \underbrace{\text{SERIALIZED INPUT } \sigma(x)}_{\text{Image / Table / Text}} \parallel \underbrace{\text{CAPTION } s(C)}_{\text{Style / Length Controlled}}, \quad (3)$$

where $\parallel$ denotes newline separation, $\sigma(\cdot)$ serializes $x$, and $s(\cdot)$ is a deterministic normalizer. Prompt templates for all benchmarks are comprehensively listed in Appendix §S2.

## 4 EXPERIMENTS

### 4.1 MAIN RESULTS

We conducted systematic experiments across 9 tasks and 9 models, covering both textual and multimodal scenarios. Further details of datasets and models are provided in Appendix S7. Each model was evaluated under two configurations, referred to as the faithful path (with a faithful description) and the hallucinated path (with a hallucinatory description). The corresponding performance results are reported in Table A1 for textual tasks and Table S4 for multimodal tasks, providing a comprehensive comparison across all datasets and models. These results allow us to distill several high-level observations that characterize how hallucinations interact with models and tasks in varied real-world scenarios. We give the following observations.

❶ **Multimodal models benefit more reliably from hallucinations.** From textual benchmarks in Table A1, we observe that only a subset of models (e.g., GPT-4o, Claude-3 Sonnet) can consistently exploit hallucinations, while others (e.g., GPT-3.5, DeepSeek-v3, Mistral Large) are often harmed. To further examine whether this trend generalizes, we extend the analysis to multimodal scenarios in Table S4. Here, the benefits of hallucinations become more pronounced: GPT-4o, Gemini-2.0-Flash, and Qwen-VL-Max show consistent improvements, with double-digit gains on perception-heavy datasets such as ISIC (up 11.8 %) and PlantVillage (up 14.7 %, up 16.9 %). This indicates hallucinations are effective in multimodal contexts, where speculative cues enrich semantic grounding of visual inputs and consistently improve downstream reasoning accuracy across tasks.

❷ **Model scale does not directly determine hallucination utility.** From Table A1, we see that larger models such as O3 and DeepSeek-R1 do not consistently benefit from hallucinations, sometimes even suffering substantial drops (e.g., drop 13.8 % on BBBP). Meanwhile, medium-sized Claude-3 Sonnet shows the largest single-task improvement (up 17.2 % on SARA). To validate this more systematically (see Appendix §S3), we further examine scaling within the Qwen2.5-VL family (Table S3). Results reveal a non-monotonic pattern: smaller (3B) and mid-sized (32B) models gain substantially, while larger variants (7B, 72B) exhibit saturated or even negative effects. This indicates that hallucination utility is governed more by architecture design and training alignment than by raw parameter count.

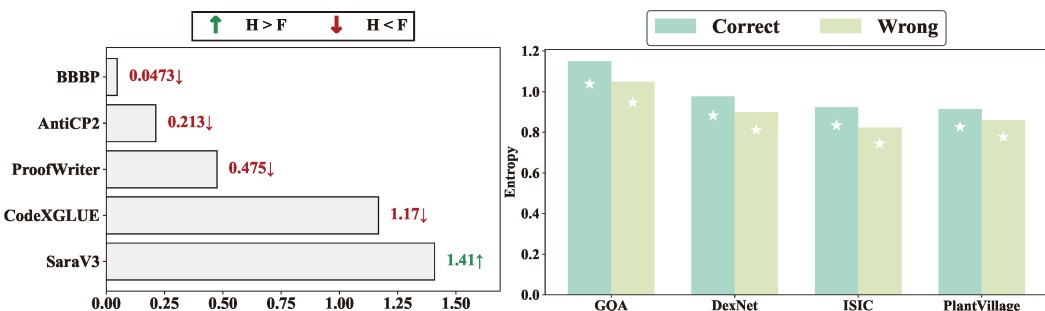

Figure 2: **Reasoning and caption entropy analysis. Left**: reasoning chain embeddings (Faithful (F) vs. Hallucinated (H)), where hallucinated prompts reshape inference trajectories with task-dependent differences ($p < 0.05$), diversifying reasoning. **Right**: caption entropy (H), where correct predictions exceed incorrect ones, confirming expanded semantic coverage. Stars indicate $p < 0.05$.

❸ **Different between perception-driven and rule-driven tasks.** From Table A1, we observe that perception-driven tasks (e.g., ProofWriter, BBBP) tend to show consistent gains from hallucinations, whereas rule-driven tasks (e.g., CodeXGLUE, SARA) often yield negligible or even negative effects. A similar pattern holds in multimodal perception benchmarks (Table S4), where vision–language tasks such as ISIC and PlantVillage exhibit significant performance gains exceeding 10% under hallucinated inputs.

❹ **Model–task interaction.** Our results suggest that hallucination effects are shaped not only by models or tasks individually, but also by their specific interaction dynamics. In particular, there may exist a potential synergy between model capability and task openness: when models with stronger hallucination–handling ability are applied to semantically open tasks (e.g., ProofWriter, ISIC), positive gains are more likely to consistently occur. By contrast, even capable models often still fail to benefit on rule-driven tasks, highlighting the inherent limits of hallucination utility.

> **Takeaway ❶.** Hallucination utility is not determined by model scale, but depends on both model design and task characteristics. It becomes beneficial primarily under a model–task synergy, where capable models align with semantically open tasks.

## 4.2 WHY HALLUCINATION HELPS

To better understand why hallucinations can be beneficial, we analyze their effects at three complementary levels: (I) **Input-level shifts**, where hallucinated captions differ significantly from faithful ones in both mean similarity and distributional spread (Fig. 3), confirming that they reshape semantic inputs rather than acting as redundant noise; (II) **Process-level modulation**, where reasoning-chain entropy analysis (Fig. 2 (left)) reveals that hallucinations alter inference dynamics—reducing entropy in some reasoning-heavy tasks (promoting convergence) while increasing it in others (support-

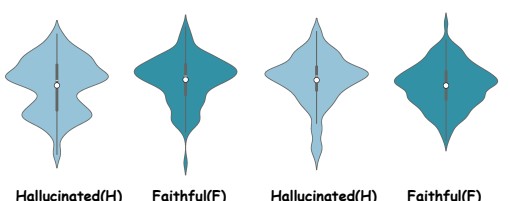

Figure 3: **Distribution of caption embeddings (Faithful (F) vs. Hallucinated (H)).** Hallucinated inputs exhibit wider semantic spread and longer tails. Stars indicate $p < 0.01$.

ing exploration); and (III) **Output-level diversity**, where correct predictions consistently exhibit higher caption entropy than incorrect ones (Fig. 2 (right)), suggesting that broader semantic coverage induced by hallucinations correlates with successful reasoning. Further implementation details of similarity computation, entropy estimation, and statistical testing are provided in Appendix §S9. We give the following observations:

❺ **Hallucinations reshape semantic inputs.** From Fig. 3, we observe that hallucinated captions differ systematically from faithful ones in both mean similarity and distributional spread. Hallucinated inputs exhibit wider variance and heavier tails in the embedding space, a difference that is

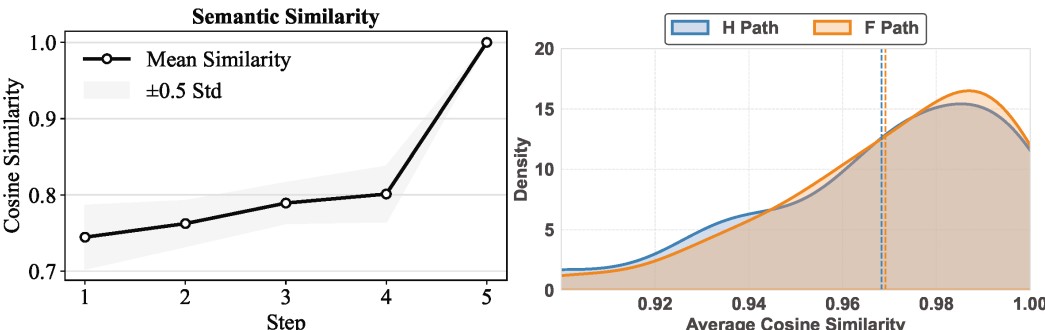

Figure 4: **Inter-chain stability on the PLANTVILLAGE dataset. Left**: Step-wise cosine similarity shows reasoning chains increasingly aligned. **Right**: Hallucinated (H) and faithful (F) captions yield overlapping similarity distributions, indicating hallucinations do not reduce reasoning stability.

statistically significant under paired t-tests ($p < 0.01$). This confirms that they introduce genuine input-level shifts rather than acting as redundant noise, providing models with additional anchors to explore alternative reasoning paths. Importantly, this shift is consistently observed across multiple datasets, underscoring that input-level semantic reshaping is a general property of hallucinations rather than a dataset-specific artifact, and holds robustly across diverse modalities and domains.

❻ **Hallucinations modulate reasoning dynamics.** As shown in Fig. 2 (left), hallucinated prompts alter the entropy of reasoning trajectories in a task-dependent manner. On reasoning-heavy tasks such as BBBP, AntiCP2, and ProofWriter, hallucinations *reduce* movement entropy, suggesting that they promote more convergent and stable inference. Conversely, on structurally open-ended tasks such as SARA-V3, hallucinations *increase* entropy, enabling the model to explore a broader range of reasoning paths. These differences are statistically significant (paired $t$-tests, $p < 0.05$), indicating that hallucinations do not merely inject noise but actively reshape inference dynamics in ways that can either encourage convergence or support exploration, depending on task demands.

❼ **Correct predictions align with higher caption entropy.** As shown in Fig. 2 (right), hallucinated captions that lead to correct predictions consistently exhibit higher semantic entropy than those leading to incorrect predictions, across datasets such as GQA, DexNet, ISIC, and PlantVillage. These differences are statistically significant ($p < 0.05$), and the effect holds consistently across all evaluated datasets, underscoring that higher semantic diversity is a general marker of successful reasoning rather than a dataset-specific artifact. Rather than mere lexical variety, this result highlights that semantic diversity in the latent space is a useful signal that supports accurate task performance.

> **Takeaway ❷.** Hallucinations consistently reshape inputs, modulate reasoning trajectories, and correlate higher semantic diversity with correct outcomes, indicating that their utility arises from broadening the semantic space rather than adding redundant noise.

### 4.3 REASONING CONVERGENCE ANALYSIS

To further examine how hallucinations influence inference stability, we analyze reasoning convergence at two complementary levels: (I) **Intra-chain convergence**, which evaluates whether intermediate reasoning steps under hallucinated captions progressively align with the final conclusion (Fig. 4 (left)); and (II) **Inter-chain consistency**, which quantifies whether different reasoning paths generated from the same input converge to semantically similar trajectories across multiple sampling seeds (Fig. 4 (right)). This two-level analysis provides a finer-grained view of convergence both within and across reasoning chains, revealing whether hallucinations promote stability or diversity in model inference. Further implementation details of similarity computation and experimental configuration are provided in Appendix §S10 to ensure clarity and reproducibility.

❽ **Hallucinations promote intra-chain convergence.** From Fig. 4 (left), we observe a clear convergence pattern: step-to-final semantic similarity steadily increases, with intermediate steps progressively aligning with the final conclusion. This rising trajectory indicates that the model's reasoning process naturally consolidates as the chain unfolds, rather than drifting away from the target answer.

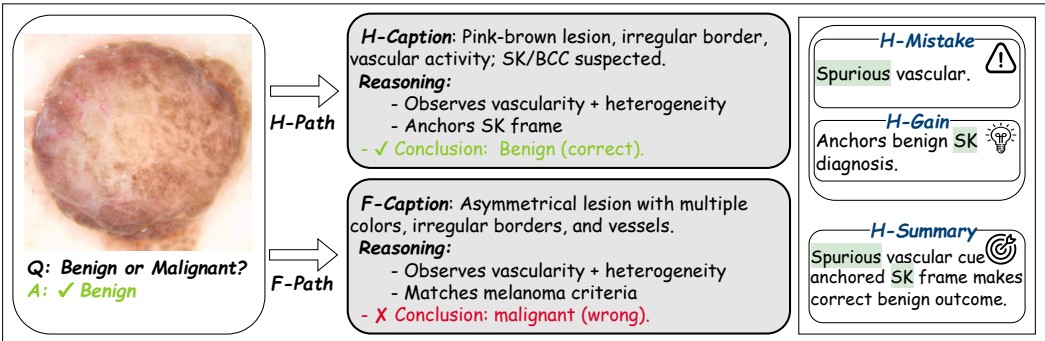

Figure 5: **Case study on ISIC.** A hallucinated (H) caption introduces a spurious vascular cue that anchors the reasoning toward a seborrheic keratosis (SK) frame, ultimately yielding the correct benign diagnosis. In contrast, the faithful (F) caption confines reasoning to superficial features, leading to a malignant misclassification, highlighting hallucination's potential as constructive guidance.

The narrowing variance band further suggests that convergence is consistent across multiple runs, reinforcing the robustness of this effect across tasks and datasets.

❾ **Reasoning chains exhibit strong inter-chain consistency.** From Fig. 4 (right), we observe that reasoning paths generated under hallucinated (H) and faithful (F) captions both achieve very high pairwise similarity across multiple sampling runs (means $\approx 0.97$). The two distributions nearly overlap, and statistical tests confirm no significant difference between them ($p > 0.6$). This indicates that, regardless of whether captions contain hallucinations, the model converges to consistent reasoning trajectories across chains. Rather than diverging into unstable alternatives, multiple sampled paths remain semantically aligned, underscoring the robustness of the model's inference.

> **Takeaway ❸.** Reasoning chains with hallucinated captions remain stable: intermediate steps consistently converge to the final answer and multiple paths sampled stay highly aligned. This stability highlights that hallucinations can support reliable and reproducible inference.

### 4.4 CASE STUDY

Beyond aggregate results, we present a case study to illustrate how hallucinations reshape reasoning chains in practice. We select a sample from the ISIC dataset, where the task is to decide whether a skin lesion is benign or malignant. Given the same input, the faithful (F) caption only describes superficial features (e.g., asymmetry, irregular borders), which confines the reasoning to melanoma criteria and leads to a misclassification as malignant. In contrast, the hallucinated (H) caption introduces a spurious vascular cue that anchors the reasoning within a seborrheic keratosis (SK) frame, ultimately guiding the model toward the correct benign diagnosis. This example highlights that hallucinations can function as semantic triggers, steering reasoning toward more effective trajectories rather than merely injecting noise. As illustrated in Fig. 5, hallucinated captions can introduce spurious but task-relevant anchors that guide the reasoning chain toward the correct outcome.

### 4.5 ABLATION STUDY

**Temperature.** We further analyze how sampling temperature influences the effect of hallucinations. As shown in Table 2, all four datasets peak at $T = 0.6$, yielding the strongest and most consistent gains (e.g., +11.76 % on ISIC, +14.68 % on PLANTVILLAGE, +2.51 % on DEXNET, +3.76 % on ANTICP2). In contrast, low temperatures ($T = 0.0/0.3$) produce overly conservative captions that truncate semantic diversity (e.g., $-4.27$ on ANTICP2, $-5.05$ on ISIC), while high temperature ($T = 0.9$) introduces excessive randomness and instability (e.g., $-5.00$ on ANTICP2). These results suggest that moderate temperature ($T = 0.6$) provides the best balance: hallucinations enrich the semantic space without overwhelming the model with irrelevant or noisy content.

Table 2: **Hallucination-induced gain ($\Delta$) across temperature and token conditions.** We report relative gain $\Delta = H - F$, isolating hallucination effects by eliminating baseline accuracy differences across datasets. **Bold** values mark the strongest gain and underlined values the second-best.

| Dataset | Temperature ($T$) | | | | Token Length | | | |
|---|---|---|---|---|---|---|---|---|
| | 0.0 | 0.3 | 0.6 | 0.9 | 128 | 256 | 512 | 1024 |
| AntiCP2 | -4.27 | +0.10 | **+3.76** | -5.00 | +0.15 | +3.76 | **+4.48** | -0.14 |
| PlantVillage | +2.30 | -4.46 | **+14.68** | +1.51 | -4.66 | **+14.68** | +7.86 | +9.49 |
| DexNet | +0.07 | +1.88 | **+2.51** | +0.14 | +1.56 | **+2.51** | -2.04 | +1.93 |
| ISIC | +9.26 | -5.05 | **+11.76** | +3.70 | -2.10 | **+11.76** | +1.33 | -0.48 |

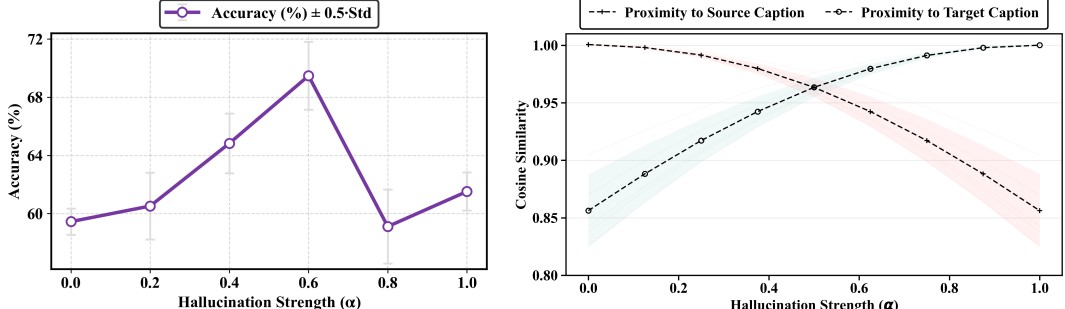

Figure 6: **Downstream task accuracy as a function of hallucination strength.** We gradually interpolate between faithful and hallucinated captions and evaluate downstream performance. The results exhibit an inverted-U pattern: introducing moderate hallucination improves accuracy, while excessive hallucination reduces it. This confirms that moderate hallucination can serve as a beneficial signal, whereas overly strong hallucination becomes detrimental.

**Maximum token budget.** We analyze the effect of token length at $T = 0.6$ (Table 2). Very short generations (128 tokens) truncate semantics and often lead to weak or negative gains. At 256 tokens, hallucinated captions deliver strong and stable improvements across datasets. Longer budgets (512/1024) sometimes yield higher peaks but with larger variance and occasional regressions. We further demonstrate the effectiveness by ablating key hallucinated tokens, which causes a substantial drop in accuracy (see Appendix §S5).

**Hallucination intensity.** To assess how hallucination strength affects performance, we generate captions with different levels of hallucination using GPT-4o and filter them into strong vs. weak groups. We also interpolate between faithful and hallucinated captions, as well as between strong and weak ones, and re-project them into SBERT space for alignment. Fig. 6 shows smooth semantic transitions (left) and an inverted-U pattern (right), where moderate hallucination improves accuracy while excessive hallucination reduces it.

## 5 Discussion and Conclusion

We distill our findings into two complementary perspectives. The *faithful path* favors exploitation: it leverages grounded evidence to produce precise but narrow predictions. The *hallucinated path* favors exploration: it introduces speculative cues that enlarge the hypothesis space and occasionally reveal useful shortcuts. Taken together, these results suggest that hallucinations are not merely errors, but alternative signals that can broaden inference. The main challenge lies in control. Hallucinations are beneficial at moderate levels, where they enrich semantics without overwhelming reliability, but they become harmful when misaligned or excessive. Effective levers include adjusting temperature and token length during generation, ensembling discriminators for stability, and using interpolation to constrain strength. Prioritizing strength scheduling and fallback to the *faithful path* helps tighten the trade-off between utility and risk. Our evaluation also highlights a crossover regime. Faithful inputs dominate in rule-driven tasks with strict correctness requirements, while hallucinatory inputs add value in semantically open or perception-heavy tasks. This points to opportunities for adaptive strategies that balance the two modes, allocating hallucination selectively to contexts where exploration improves outcomes and suppressing it where stability is paramount.

ETHICS STATEMENT

We conform to the ICLR Code of Ethics and provide the asset license and consent information in Appendix §S12. All datasets used in this study are publicly available benchmarks with clearly specified licenses (*e.g.*, GPL-3.0, MIT, Apache 2.0, CC BY 4.0, CC BY-NC; see Appendix §S12). All models are also publicly available APIs or checkpoints released by their respective providers, governed by open-source licenses (e.g., Apache 2.0, Qwen Research License) or official API service terms. We emphasize that our use of both datasets and models is strictly for academic research purposes, in accordance with their license conditions. The datasets may contain biomedical, legal, or other sensitive content, but such content does not represent the views of the authors. Our work does not involve crowdsourcing or human-subject studies.

REPRODUCIBILITY STATEMENT

All experiments in this paper are evaluation-only. Our implementation is based on PyTorch (Paszke et al., 2019) and runs on NVIDIA RTX 4090 GPUs. We evaluate publicly available models on publicly available datasets (see Appendix §S12 for details). We provide the exact datasets and evaluation metrics in Appendix §S7, enabling reproduction of our reported results. Our evaluation scripts will be released upon acceptance.

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

## SUMMARY OF THE APPENDIX

This supplementary contains additional details for fourteenth International Conference on Learning Representations submission, titled *"Productive LLM Hallucinations: Conditions, Mechanisms, and Benefits"*. The supplementary is organized as follows:

- §S1 reports the **significance and robustness analysis**. It includes mean±std, $\Delta$(H–F), and $p$-values across datasets.

- §S2 lists the **prompt templates**. Each dataset has a role prompt, a generation prompt, and an evaluation prompt.

- §S3 presents the **effect of model scale**. Hallucination gains are non-monotonic across Qwen2.5-VL sizes.

- §S4 presents the **random checker control**. It confirms that improvements are not due to arbitrary filtering..

- §S5 reports the **token ablation study**. Core hallucinated tokens are shown to be necessary for success.

- §S6 presents the **case studies**. It provides qualitative examples on DexNet, BBBP, and PlantVillage, showing how hallucinated captions act as anchors that guide reasoning toward correct outcomes

- §S7 describes the **experimental setup**. It includes datasets, models, and evaluation protocols.

- §S8 explains the **caption discriminator**. Three complementary factuality verifiers are described in detail.

- §S9 provides the **analysis setup**. Input-level, process-level, and output-level analysis pipelines are described separately.

- §S10 provides the **convergence and similarity analysis**. Both intra-chain and inter-chain convergence are reported.

- §S11 evaluates the **HIVE discriminator performance**. Accuracy is reported on TruthfulQA and curated datasets.

- §S12 summarizes the **dataset licenses**. It also summarizes the **model licenses**.

- §S13 provides the **AI disclosure**. GPT-5 was used only for grammar checking.

## S1 SIGNIFICANCE AND ROBUSTNESS

Table S1: **Statistical significance of hallucination-induced gains.** Mean±Std over 5 runs. $\Delta$(H–F) denotes accuracy gain. Two-sided paired $t$-test $p$-values; significant results ($p<0.05$) are bolded.

| Dataset | Domain | Faithful (F) | Hallucinated (H) | $\Delta$(H–F) | $p$ |
|---|---|---|---|---|---|
| AntiCP2 | Protein | 0.5459±0.0067 | 0.5835±0.0126 | +0.0376 | **0.00036** |
| BBBP | Drug property | 0.6167±0.0264 | 0.6833±0.0118 | +0.0667 | **0.00481** |
| CodeXGLUE | C++ code | 0.5515±0.0326 | 0.5275±0.0227 | −0.0240 | 0.102 |
| SARA_V3 | Law reasoning | 0.6293±0.0084 | 0.6224±0.0161 | −0.0069 | 0.147 |
| ProofWriter | Logic | 0.6949±0.0125 | 0.7500±0.0217 | +0.0551 | $1.57\times10^{-5}$ |
| GQA | VQA multimodal | 0.7136±0.0040 | 0.7422±0.0078 | +0.0286 | **0.00088** |
| DexNet | Robotics | 0.5325±0.0045 | 0.5576±0.0116 | +0.0251 | **0.00038** |
| ISIC | Medical | 0.6388±0.0126 | 0.7564±0.0248 | +0.1176 | **0.00151** |
| PlantVillage | Agriculture | 0.6273±0.0249 | 0.7741±0.0211 | +0.1468 | $3.42\times10^{-9}$ |

All reported results are averaged over five independent runs with different random seeds and presented as mean±std. We conduct two-sided paired $t$-tests to compare faithful (F) and hallucinated (H) inputs. Table S1 reports the full results across all nine datasets, including mean±std, relative gain $\Delta$(H–F), and $p$-values. The majority of tasks exhibit statistically significant gains ($p<0.05$

Table S2: Prompts used across datasets. Each dataset is paired with a role prompt, a generation prompt, and an evaluation prompt, ensuring task-specific context and consistency.

| Dataset | Role prompt | Generation prompt | Evaluation prompt |
|---|---|---|---|
| AntiCP2 | You are a protein science expert. | {Sign}\n Describe this in natural language: | You are a peptide bioinformatics expert responsible for evaluating short peptide sequences for the presence or absence of anticancer activity. Answer: yes or no. Then provide a step-by-step reasoning process. |
| BBBP | You are an expert in drug discovery. | {Sign}\n Describe this in natural language: | Does the molecule have the ability to penetrate the blood-brain barrier? Answer: yes or no. Then provide a step-by-step reasoning process. |
| CodeXGLUE | You are a software security expert and professor. | {Sign}\n Describe this in natural language: | You are a software security expert and professor. Does the following C function contain a security vulnerability? Answer: yes or no. Then provide a step-by-step reasoning process. |
| SARA_V3 | You are a legal expert. | {Sign}\n Describe this in natural language: | You are a legal reasoning assistant. Determine whether the following legal claim is supported by the facts. Answer: yes or no. Then provide a step-by-step reasoning process. |
| Proof | You are an assistant for reasoning. | {Sign}\n Describe this in natural language: | You are a logical reasoning assistant. Determine whether the statement is entailed by the given context of facts and rules. Answer: yes or no. Then provide a step-by-step reasoning process. |
| GQA | You are a reason expert. | Describe this image in natural language: | You are a visual reasoning expert. Answer the question based on the image. Answer: yes or no. Then provide a step-by-step reasoning process. |
| Dex_Net | You are an expert in robotic grasp assessment. | Describe this image in natural language: | You are a senior robotic manipulation engineer specializing in parallel-jaw grasp planning. Answer: yes or no. Then provide a step-by-step reasoning process. |
| ISIC | You are an expert dermatoscopist. | Describe this image in natural language: | You are an expert dermatoscopist. Based on this image, decide whether the lesion is malignant (melanoma) or benign. Answer: yes (malignant melanoma) or no (benign). Then provide a step-by-step reasoning process. |
| PlantVillage | You are a seasoned plant pathologist for solanaceous crops | Describe this image in natural language: | You are an expert plant pathologist who diagnoses tomato foliar diseases. Decide whether it shows early blight or late blight: reply yes if it is early blight and no if it is late blight. Answer: yes or no. Then provide a step-by-step reasoning process. |

or $p<0.01$). For example, hallucinations yield large and consistent improvements on perception-heavy datasets such as ISIC ($+11.8\%$, $p = 0.0015$) and PlantVillage ($+14.7\%$, $p<10^{-8}$), while rule-driven tasks such as CodeXGLUE and SARA show negligible or non-significant differences ($p>0.1$). These results confirm that the reported improvements are statistically reliable rather than random variation.

## S2 PROMPT TEMPLATES

To ensure consistency across benchmarks, we design unified prompt templates that follow a three-part structure: a *role prompt*, a *generation prompt*, and an *evaluation prompt*. The role prompt assigns the model an expert identity tailored to the domain (e.g., drug discovery, legal reasoning, medical diagnosis). The generation prompt asks the model to verbalize the raw input ({Sign}) into natural language, thereby producing either a faithful or a hallucinated caption. Finally, the evaluation prompt specifies the downstream task, which always requires a binary decision (yes/no) together with a step-by-step reasoning chain. This design ensures that the only experimental variable is the type of caption (faithful vs. hallucinated), while all other aspects of the prompt remain controlled.

Table S2 lists the complete templates used for all nine datasets. These include both text-based tasks (AntiCP2, BBBP, CodeXGLUE, SARA, ProofWriter) and multimodal tasks (GQA, DexNet, ISIC, PlantVillage). The templates were fixed across all models and experiments, so that observed differences can be attributed solely to the presence or absence of hallucinated semantics.

## S3 EFFECT OF MODEL SCALE

We evaluate Qwen2.5-VL models at four different scales (3B, 7B, 32B, 72B). As shown in Table S3, the impact of hallucinations is not monotonic with scale. Both 3B and 32B models benefit substantially (+9.4 %), while the 7B and 72B models show slight drops. This suggests that scale alone does not determine hallucination effectiveness: smaller models may gain from additional semantic cues, whereas very large models may already saturate on faithful inputs, making further hallucinations redundant or even distracting.

Table S3: Scaling results within the Qwen2.5-VL family. Hallucination effects are non-monotonic: smaller and medium-large models benefit, while very large models show volatility or saturation.

| Model | Faithful (F) | Hallucinated (H) | $\Delta$(H-F) |
|---|---|---|---|
| Qwen2.5-VL-3B | 0.5935±0.0068 | **0.6874±0.0409** | +0.0939 |
| Qwen2.5-VL-7B | 0.5898±0.0000 | 0.5276±0.0000 | −0.0622 |
| Qwen2.5-VL-32B | 0.5935±0.0068 | **0.6874±0.0409** | +0.0939 |
| Qwen2.5-VL-72B | **0.7349±0.0171** | 0.6856±0.0167 | −0.0493 |

Table S4: **Faithful vs. Hallucinated accuracy (Image+Question).** Cells show mean accuracy (%). $\Delta$ denotes H–F (%) and is shown inline at bottom-right.

| Dataset | P. | GPT-4o | Claude-3 Sonnet | Gemini-2.0 Flash | Qwen VL-Max |
|---|---|---|---|---|---|
| GQA | F | $71.36_{(-)}$ | $62.02_{(-)}$ | $75.23_{(-)}$ | $69.88_{(-)}$ |
|  | H | $74.22_{\uparrow+2.86}$ | $61.03_{\downarrow-0.99}$ | $75.69_{\uparrow+0.46}$ | $67.90_{\downarrow-1.98}$ |
| DexNet | F | $53.25_{(-)}$ | $50.29_{(-)}$ | $49.88_{(-)}$ | $51.54_{(-)}$ |
|  | H | $55.76_{\uparrow+2.51}$ | $50.71_{\uparrow+0.42}$ | $51.15_{\uparrow+1.27}$ | $54.12_{\uparrow+2.58}$ |
| ISIC | F | $63.88_{(-)}$ | $54.19_{(-)}$ | $67.23_{(-)}$ | $58.71_{(-)}$ |
|  | H | $75.64_{\uparrow+11.76}$ | $61.02_{\uparrow+6.83}$ | $67.90_{\uparrow+0.67}$ | $75.62_{\uparrow+16.91}$ |
| PlantVillage | F | $62.73_{(-)}$ | $55.28_{(-)}$ | $62.71_{(-)}$ | $67.84_{(-)}$ |
|  | H | $77.41_{\uparrow+14.68}$ | $72.50_{\uparrow+17.22}$ | $70.98_{\uparrow+8.27}$ | $77.66_{\uparrow+9.82}$ |

## S4 RANDOM CHECKER CONTROL

To rule out improvements arising from arbitrary filtering, we replace our hallucination checker with a *random* checker that accepts hallucinations without factual assessment. All decoding controls are kept fixed. As shown in Table S5, random filtering yields only negligible gains (0.17–1.23 %) and no statistical significance on any dataset (all $p > 0.14$), indicating that the benefits reported in the main results do not stem from chance.

Table S5: **Random checker ablation.** Mean±std over 5 runs. $\Delta$ denotes the absolute accuracy difference (H–F). $p$ from two-sided paired $t$-tests; (n.s.) = not significant at $p<0.05$.

| Dataset | Faithful (F) | H + Random (H) | $\Delta$ (H–F) | $p$ |
|---|---|---|---|---|
| AntiCP2 | $0.5015 \pm 0.0124$ | $0.5114 \pm 0.0200$ | +0.0100 | 0.526 (n.s.) |
| PlantVillage | $0.7358 \pm 0.0189$ | $0.7481 \pm 0.0205$ | +0.0123 | 0.354 (n.s.) |
| DeXNet | $0.4950 \pm 0.0068$ | $0.4967 \pm 0.0062$ | +0.0017 | 0.757 (n.s.) |
| ISIC | $0.5763 \pm 0.0067$ | $0.5805 \pm 0.0042$ | +0.0043 | 0.142 (n.s.) |

## S5 TOKEN ABLATION: NECESSITY OF HALLUCINATED EVIDENCE

We conduct a token-level ablation to test whether hallucinated content is *necessary* for success. Concretely, we (i) filter to the subset of samples that are solvable only with hallucinated captions

(*H-before* succeeds while faithful captions fail), (ii) identify the hallucinated tokens that serve as *core evidence* in the model's reasoning, and (iii) mask these tokens in the hallucinated captions (using a neutral placeholder) and re-evaluate on the same samples. Table S6 reports post-ablation accuracy (*H-after, ablated*) across four datasets. The sizable drops from near-perfect *H-before* (not shown here for brevity) to the post-ablation scores demonstrate that core hallucinated tokens are not noise, but carry information the model *relies on* to solve the tasks.

Table S6: **Token ablation on hallucinated captions.** Accuracy after masking hallucinated tokens that were used as *core evidence* in the model's reasoning (evaluated only on the subset solvable by hallucinated captions).

| Dataset | H-after (ablated) Acc. |
|---|---|
| AntiCP2 | $0.244 \pm 0.085$ |
| PlantVillage | $0.700 \pm 0.111$ |
| DexNet | $0.364 \pm 0.061$ |
| ISIC | $0.380 \pm 0.062$ |

## S6 QUALITATIVE CASE STUDIES

**DexNet: Robotic Grasping.** Fig. S2 shows a robotic grasping case from DexNet. The hallucinated (H) caption mistakenly interprets the depth map as a wheeled robot silhouette, but this spurious cue provides a concrete object anchor that enables correct reasoning for graspability. In contrast, the faithful (F) caption only describes gray gradients and claw-like shapes, failing to establish object identity and thus leading to the wrong "No" prediction. This case illustrates how hallucinations, though factually incorrect, can enrich the reasoning space and support the correct decision.

**BBBP: Molecular Permeability.** Fig. S3 presents a molecular classification example from BBBP. The hallucinated (H) caption misidentifies the scaffold as naphthalene, but this cue anchors reasoning toward favorable hydrophobicity, guiding the model to correctly predict blood–brain barrier penetration. Meanwhile, the faithful (F) caption emphasizes a biphenyl scaffold with protonated amine, anchoring reasoning on size/charge constraints and resulting in the wrong "No" prediction. This case demonstrates that even erroneous aromatic anchors can serve as constructive signals for correct permeability classification.

**PlantVillage: Crop Disease Recognition.** Fig. S1 illustrates a crop disease recognition task. The hallucinated (H) caption highlights darkened tips and edges, anchoring reasoning toward late blight and producing the correct diagnosis. In contrast, the faithful (F) caption notes similar edge darkening but does not explicitly anchor late blight, leading to the incorrect early-blight decision. This case underscores that hallucinations, even when based on spurious cues, can act as decisive anchors that steer reasoning toward the correct outcome.

**Summary of Case Study.** Across DexNet, BBBP, and PlantVillage, a consistent pattern emerges: hallucinated captions often introduce spurious or factually mistaken cues (e.g., a robot silhouette, a naphthalene core, or darkened leaf edges). Yet these cues act as decisive anchors that expand the reasoning space, providing additional structure that guides the model toward the correct outcome. By contrast, faithful captions though factually accurate may lack sufficient anchoring, causing reasoning to remain shallow and sometimes incorrect. These case studies highlight hallucination's constructive potential: even when imperfect, hallucinations can inject inductive signals that improve decision quality.

## S7 EXPERIMENTAL SETUP

**Datasets.** We conduct experiments on 9 datasets spanning both textual and multimodal domains. *Text (5):* AntiCP2 (Agrawal et al., 2021) (antimicrobial peptide classification), BBBP (Martins et al., 2012) (blood–brain barrier penetration), CodeXGLUE (Lu et al., 2021) (C++ exception prediction), SARA (Henderson et al., 2022) (legal reasoning), ProofWriter (Tafjord et al., 2021) (logic-based

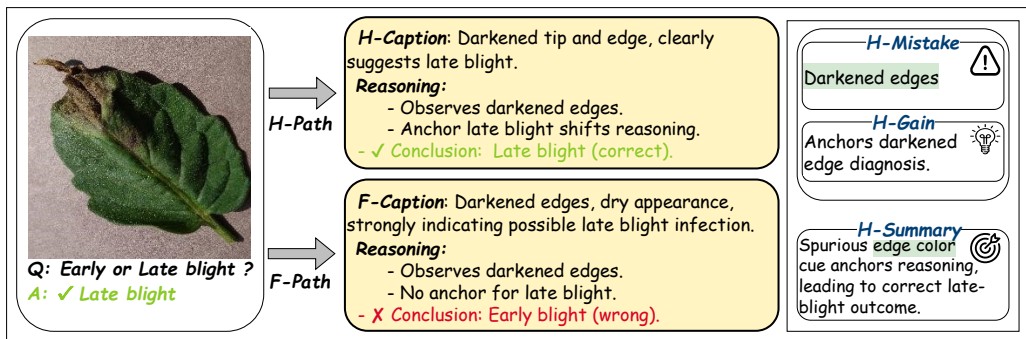

Figure S1: **Case study on PlantVillage.** A hallucinated (H) caption highlights a spurious cue darkened tip and edges that anchors reasoning toward late blight, ultimately yielding the correct diagnosis. In contrast, the faithful (F) caption notes the same darkened edges but lacks an explicit anchor for late blight, leading to an incorrect early-blight classification. This example illustrates how hallucinations, even when grounded in partially misleading features, can provide decisive anchors that guide reasoning toward the correct outcome.

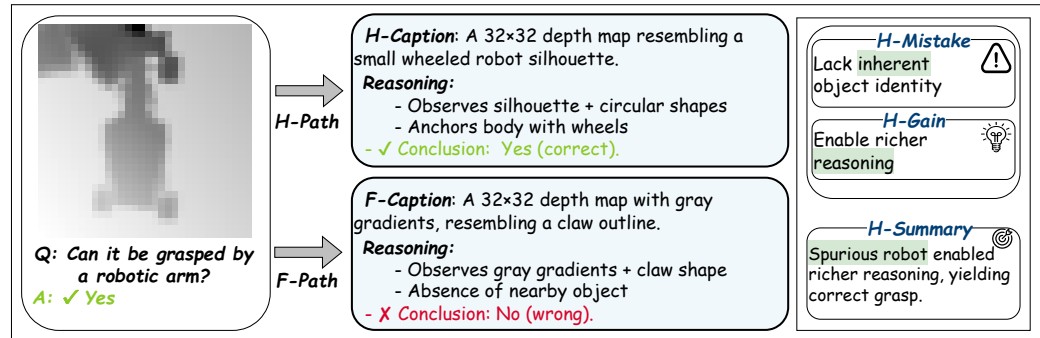

Figure S2: **Case study on DexNet.** A hallucinated (H) caption misinterprets the depth map as a robot silhouette with wheels, anchoring reasoning toward a graspable object and yielding the correct answer. In contrast, the faithful (F) caption only notes gray gradients and claw-like shapes, failing to establish object identity and leading to an incorrect "No" prediction. This example shows how hallucinated cues, though factually incorrect, can enrich reasoning and enable correct decisions.

natural language inference). *Multimodal (4):* GQA (Hudson & Manning, 2019) (visual question answering), DexNet (Mahler et al., 2017) (depth-based robotic grasping), ISIC (Tschandl et al., 2018) (skin-lesion classification), PlantVillage (Hughes et al., 2015) (plant-disease recognition from RGB images).

**Models.** We evaluate 9 large language models, covering both proprietary and open-source systems:
*Closed-source:* GPT-4o, GPT-3.5-turbo, Claude 3 Sonnet, Gemini 2.0 Flash, O3.
*Open-source:* DeepSeek-V3, DeepSeek-R1, Mistral Large, Qwen-VL

**Evaluation.** For binary classification tasks, we report Accuracy as the primary metric, ensuring consistency across datasets and model families.

**Statistics.** Unless otherwise noted, we report mean±std over 5 independent runs. For each dataset, we conduct two-sided paired *t*-tests to compare faithful vs. hallucinated inputs. Statistical significance is reported at conventional thresholds ($p<0.01$); full results and additional details are provided in Appendix S1. Token budget is implemented as a maximum generation length, though generations may terminate earlier.

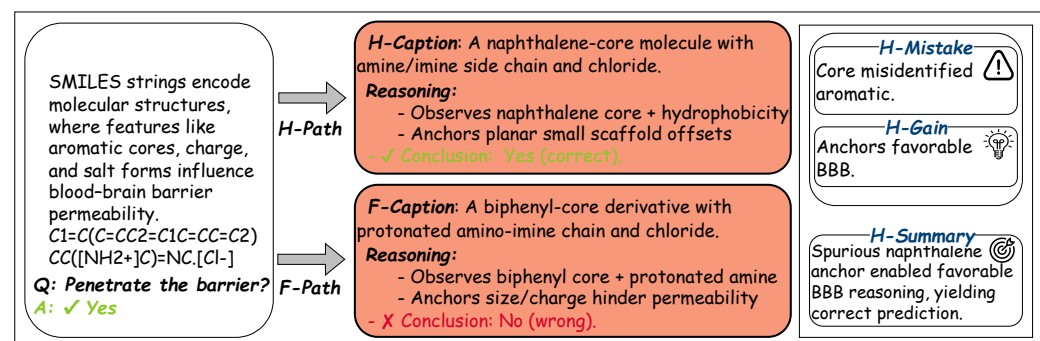

Figure S3: **Case study on BBBP.** A hallucinated (H) caption incorrectly identifies the molecule as naphthalene-based but introduces a hydrophobic anchor favoring blood–brain barrier permeability, leading to the correct Yes" outcome. In contrast, the faithful (F) caption focuses on a biphenyl scaffold with protonated amine, anchoring reasoning on size and charge constraints and resulting in the wrong No" prediction. This example illustrates how aromatic cues, though factually mistaken, can provide constructive anchors that guide reasoning toward correct molecular permeability.

## S8 IMPLEMENTATION OF CAPTION DISCRIMINATOR

We implement three complementary modules to assess the factual plausibility of hallucinated captions. Each module is motivated by prior work on self-consistency, fine-grained fact-checking, and paraphrase-based semantic validation.

**Fine-Grained Factuality Verifier.** Motivated by the fine-grained evaluation perspective of Factcheck-Bench (Wang et al., 2024), this module decomposes each caption into individual factual claims using sentence segmentation. Each claim is independently verified by a large language model with a structured prompt that returns a binary verdict (True/False), a confidence score (0–1), and a short justification. The final score averages the confidence of verified True claims, with a penalty for detected False claims. This design enables auditing hallucinations at the individual claim level, rather than only at the aggregated whole-caption level used previously in prior evaluations.

**Self-Evaluation Factuality Verifier.** Inspired by self-consistency approaches such as SelfCheck-GPT (Manakul et al., 2023), this module prompts the model to directly self-assess the factual correctness of an answer (with optional question context) in multiple practical scenarios. The model outputs a binary verdict with confidence and explanation. Compared to the fine-grained verifier, this method is lightweight and evaluates factuality at the whole-answer level. We also support a multimodal variant that incorporates image inputs when available across diverse evaluation settings.

**Paraphrase-Consistency Verifier.** Following the idea of leveraging paraphrasing and question generation for semantic consistency (Liu et al., 2020), this module generates two paraphrases of the original caption while strictly preserving meaning. The paraphrases serve only as auxiliary evidence to clarify intent, while the factuality decision always prioritizes the original caption. A fact-adjudication prompt then produces a binary verdict, confidence, and concise reasoning. This consistency check reduces prompt variance and stabilizes factuality judgments overall accuracy.

**Caption Discriminator.** Together, these three discriminators provide complementary perspectives: (I) fine-grained claim verification, (II) holistic self-evaluation, and (III) paraphrase-assisted consistency. In our experiments, we include a *random checker* baseline (accept/reject uniformly at random) and ensemble variants combining multiple verifiers. Results in Appendix §S4 confirm that random filtering yields no significant gains, while learned discriminators provide stable improvements.

## S9 IMPLEMENTATION DETAILS OF ANALYSIS SETUP

**Input-level.** To quantify the differences between faithful and hallucinated captions, and between correct and incorrect predictions, we adopt the following analysis pipeline. For each dataset, we collect hallucinated caption embeddings produced by the generation model. When multiple runs are available, we resolve embeddings by searching run-specific directories or aggregated to ensure

consistent coverage. Captions and embeddings are aligned with prediction labels, with samples truncated if necessary to guarantee matching length. High-dimensional embeddings are projected to a three-dimensional latent space using principal component analysis (PCA) with full SVD. This preserves the dominant semantic directions while removing redundant variance, facilitating density estimation. We estimate local distributional entropy of the embeddings by fitting a Gaussian kernel density estimator (KDE) with fixed bandwidth. For each sample, the negative log-density serves as its entropy value, reflecting whether it lies in a dense or sparse region of the semantic space. We report mean and standard deviation of entropy separately for correct and incorrect predictions. To assess the significance of differences between correct and incorrect groups, we conduct two-sided independent-sample $t$-tests without assuming equal variance. We report the test statistic and $p$-value for each dataset. Results are aggregated across all nine benchmarks and summarized in Appendix §S1. This procedure provides a principled way to examine how hallucinations reshape semantic distributions at the input level, modulate reasoning trajectories, and correlate with prediction accuracy through entropy-based analysis.

**Process-level.** To examine how hallucinations modulate inference dynamics, we quantify the entropy of reasoning-chain embeddings. For each input, we record the hidden-state representations of step-wise reasoning trajectories under both faithful (F) and hallucinated (H) captions. We then project these embeddings into a lower-dimensional space via principal component analysis (PCA) and estimate their density distribution using kernel density estimation (KDE). The negative log-likelihood of KDE outputs serves as an entropy measure, capturing the dispersion of reasoning movements across steps. For each dataset, we compute the mean entropy under F and H conditions, and report their differences as shown in Fig. 2 (left). Paired two-sided $t$-tests are applied to assess statistical significance ($p < 0.05$). This measurement allows us to characterize whether hallucinations encourage more convergent reasoning trajectories (lower entropy) or diversify inference paths (higher entropy), depending on the task structure.

**Output-level.** To analyze the semantic effect of hallucinations, we estimate the entropy of caption embeddings under the hallucinated (H) condition and compare between correct and incorrect predictions. For each dataset, we collect the OpenCLIP embeddings of hallucinated captions ($C_H$). Predictions and gold labels are aligned with these embeddings by matching the number of instances. To improve stability and reduce noise in density estimation, embeddings are projected into a 3-dimensional latent space using Principal Component Analysis (PCA). This preserves the dominant variance directions while mitigating the curse of dimensionality. We adopt Kernel Density Estimation (KDE) with a Gaussian kernel (bandwidth = 0.5) to approximate the underlying semantic distribution. For each sample, we compute the negative log-likelihood under the KDE as a proxy for semantic entropy. We split samples into two groups based on prediction correctness and compute mean $\pm$ standard deviation of entropy for each group. Statistical differences are assessed using two-sided $t$-tests under unequal variance assumptions. This procedure yields a robust measure of semantic diversity in hallucinated captions, allowing us to test whether correct predictions are associated with higher entropy than incorrect. Table S7 summarizes caption entropy under hallucinated (H) inputs, split by correct vs. incorrect predictions. We find that correct predictions generally align with higher entropy, with significant differences on four multimodal datasets (GQA, DexNet, ISIC, PlantVillage). These results confirm that semantic diversity is a reliable marker of successful reasoning rather than a dataset-specific artifact.

## S10 IMPLEMENTATION DETAILS OF CONVERGENCE AND SIMILARITY ANALYSIS

**Intra-chain convergence.** To further understand the internal dynamics of hallucinated reasoning, we analyze whether intermediate steps in a reasoning chain progressively converge toward the final conclusion. Specifically, we extract step-wise reasoning traces from hallucinated captions and compute semantic embeddings using OpenCLIP (ViT-L/14, OpenAI weights). Each intermediate step is compared to the final step via cosine similarity, yielding a *step-to-final similarity curve* averaged across reasoning chains. As shown in Fig. 4 (left), similarity consistently increases as the chain progresses, while variance bands narrow, indicating that hallucinated reasoning exhibits stable intra-chain convergence. This suggests that intermediate steps are not drifting away but instead steadily aligning with the final conclusion.

Table S7: **Caption entropy analysis (H condition).** Entropy compared between correct and wrong predictions. Values show mean entropy for each group, their difference, and two-sided $t$-tests. Significant results ($p < 0.05$) are bolded.

| Dataset | Correct | Wrong | $\Delta$(C-W) | $t$-stat | $p$-value |
|---|---|---|---|---|---|
| AntiCP2 | 0.861 | 0.856 | $+0.005$ | 0.32 | 0.749 |
| BBBP | 0.886 | 0.960 | $-0.074$ | $-2.26$ | 0.032 |
| CodeXGLUE | 0.901 | 0.897 | $+0.004$ | 0.30 | 0.768 |
| SARA_V3 | 1.030 | 1.004 | $+0.025$ | 1.91 | 0.060 |
| ProofWriter | 0.975 | 0.998 | $-0.023$ | $-1.14$ | 0.262 |
| GQA | 1.150 | 1.048 | $+0.102$ | 4.61 | $1.4\times10^{-5}$ |
| DexNet | 0.977 | 0.899 | $+0.078$ | 2.76 | **0.006** |
| ISIC | 0.923 | 0.823 | $+0.100$ | 3.87 | **0.0012** |
| PlantVillage | 0.914 | 0.860 | $+0.054$ | 3.05 | **0.0030** |

**Inter-chain convergence.** To further evaluate the stability of reasoning trajectories, we computed the *average path similarity* across multiple sampled chains. For each dataset, we first extracted hallucinated (H) and non-hallucinated (NH) reasoning paths, then embedded all intermediate steps using OpenCLIP (ViT-L/14, OpenAI weights). The cosine similarity between different runs was averaged to yield an overall *path-level similarity score*. We then compared the distribution of average similarities between H and NH conditions. Kernel density estimation (KDE) was applied to visualize the distributions, as shown in Fig. 4 (right). Results indicate that both H and NH paths consistently achieve very high similarity (means $\approx 0.97$), with nearly overlapping distributions. This confirms that hallucinations do not compromise inter-chain stability, and that multiple reasoning paths remain semantically aligned across runs.

## S11 HIVE PERFORMANCE EXPERIMENT

To assess the reliability of HIVE's hallucination discriminator, we evaluate it in two settings. First, on the TruthfulQA benchmark, which is commonly used to probe hallucination, the discriminator achieves 81.76% accuracy. Second, to approximate real-world, cross-domain use, we curate a 180-sample dataset by sampling 20 captions from each of nine tasks, manually annotate them as either hallucination or faithful, and back-test the discriminator; it attains 83.72% accuracy. These results demonstrate that the module generalizes beyond a single benchmark and effectively separates hallucinated from faithful captions, providing a stable foundation for subsequent comparisons.

## S12 LICENSE

**Datasets.** All datasets used in this study are publicly available benchmarks. Their license terms are as follows: AntiCP2 is released under GPL-3.0; BBBP under the MIT License; CodeXGLUE under the Computational Use of Data Agreement (C-UDA); SARA_V3 under CC BY 4.0; ProofWriter under CC BY 4.0; GQA annotations under CC BY 4.0; Dex-Net code under BSD-3-Clause while its HDF5 databases are restricted to research-only (non-commercial) use; ISIC under CC BY-NC (non-commercial); and PlantVillage under CC0. We emphasize that our use of these datasets is strictly for academic research purposes.

**Models.** All models used in this study are publicly available APIs or checkpoints released by their respective providers. Specifically, Qwen2.5-VL-3B and Qwen2.5-VL-72B are released under the Qwen Research License, while Qwen2.5-VL-7B and Qwen2.5-VL-32B adopt the Apache 2.0 License. For commercial API models, including GPT-4o, GPT-3.5-turbo (OpenAI), Claude 3 Sonnet (Anthropic), Gemini 2.0 Flash (Google DeepMind), O3 (OpenAI), DeepSeek-V3 and DeepSeek-R1 (DeepSeek), Mistral Large (Mistral), and Qwen-VL (Alibaba), usage is governed by their providers' service terms and API agreements. We emphasize that our use of these models is strictly for academic research purposes in accordance with their public availability and license terms.

## S13    AI DISCLOSURE

We acknowledge the use of GPT-5 for grammar checking only. The model was employed to correct grammatical errors while ensuring the original meaning and intent of the text remained unchanged.

# APPENDIX A. APPENDIX FOR REBUTTAL

## A1 TABLE UPDATE

Here we provide the corrected version of Table 1 referenced in the rebuttal.

Table A1: **Faithful (F) vs. Hallucinated (H) path accuracy.** Denote $\Delta$(H–F) as the relative accuracy performance gain ↑ or drop ↓ from the hallucinated path over the faithful path.

| Dataset | P. | GPT-4o | GPT-3.5 | Claude-3 Sonnet | DeepSeek v3 | Mistral Large | O3 | DeepSeek R1 |
|---|---|---|---|---|---|---|---|---|
| AntiCP2 | F | 54.59(−) | 43.63(−) | 46.84(−) | 52.19(−) | 56.69(−) | 53.95(−) | 49.87(−) |
| | H | 58.35↑3.76 | 47.76↑4.13 | 48.21↑1.37 | 45.01↓-7.18 | 57.84↑1.15 | 50.17↓-3.78 | 46.42↓-3.45 |
| BBBP | F | 61.67(−) | 60.75(−) | 64.07(−) | 61.60(−) | 59.41(−) | 73.27(−) | 70.53(−) |
| | H | 68.33↑6.66 | 57.53↓-3.22 | 64.95↑0.88 | 55.56↓-6.04 | 59.65↑0.24 | 59.47↓-13.80 | 58.88↓-11.65 |
| CodeXGLUE | F | 55.15(−) | 58.90(−) | 49.25(−) | 49.46(−) | 50.11(−) | 45.10(−) | 51.54(−) |
| | H | 52.75↓-2.40 | 57.40↓-1.50 | 53.40↑4.15 | 50.13↑0.67 | 56.40↑6.29 | 46.06↑0.96 | 48.95↓-2.59 |
| SARA | F | 62.93(−) | 52.28(−) | 62.07(−) | 58.10(−) | 59.14(−) | 65.52(−) | 58.62(−) |
| | H | 62.24↓-0.69 | 54.83↑2.55 | 63.97↑1.9 | 58.96↑0.86 | 60.34↑1.20 | 62.07↓-3.45 | 54.48↓-4.14 |
| ProofWriter | F | 69.49(−) | 66.55(−) | 76.03(−) | 85.17(−) | 70.86(−) | 97.76(−) | 89.31(−) |
| | H | 75.00↑5.51 | 66.21↓-0.34 | 76.73↑0.70 | 85.86↑0.69 | 68.62↓-2.24 | 98.45↑0.69 | 92.93↑3.62 |

## A2 FIGURE UPDATE

Here we provide the updated version of Figure 4 referenced in the rebuttal. The revision improves clarity without affecting any results or conclusions.

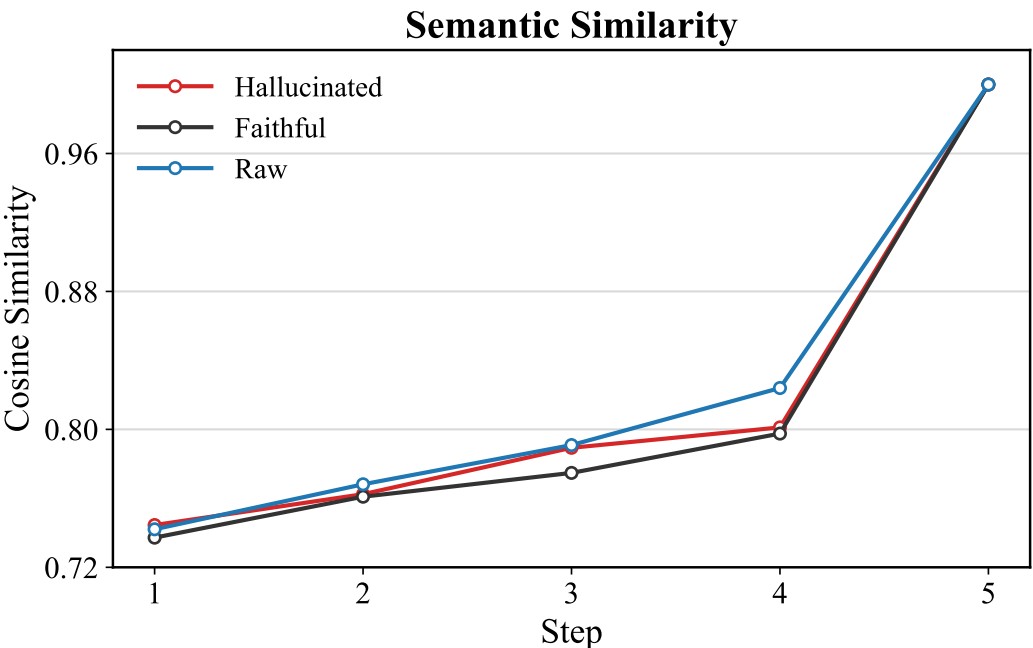

Figure A1: Updated Figure 4(left). Step-wise cosine similarity of the reasoning chain under Raw, Faithful (F), and Hallucinated (H) inputs. All three curves exhibit a similar monotonic convergence toward the final step, indicating that hallucinated captions do not disrupt the chain structure. The updated visualization improves clarity without changing any results or interpretations.

