# OpenReview forum: "Productive LLM Hallucinations: Conditions, Mechanisms, and Benefits"
_ICLR.cc/2026/Conference — Submitted to ICLR 2026_

### Official Review · Reviewer_fxuZ · 2025-10-22

**Soundness:** 3
**Presentation:** 2
**Contribution:** 3
**Rating:** 6
**Confidence:** 3

**Summary:**

This paper explores the counterintuitive phenomenon that hallucinated inputs can, under certain conditions, enhance model performance across a range of tasks and architectures. The authors introduce HIVE, a systematic framework designed to evaluate the impact of hallucinations on downstream task performance. Using this framework, they generate both hallucinated and faithful captions for the same input and quantify the performance gap between two settings. The analysis spans both textual and multimodal tasks and involves multiple large language models. Empirical results demonstrate that hallucinated captions consistently improve accuracy on multimodal and perception-driven tasks. These findings provide novel insights into the productive role of hallucinations and open promising directions for leveraging controlled hallucination to enhance LLM reasoning capabilities.

**Strengths:**

- The proposed evaluation framework is conceptionally simple and task-agnostic
- The authors conduct comprehensive evaluation across diverse datasets, with significance testing and robustness checks, to validate the effect of hallucinations on model performance
- The paper is well structured and easy to follow. The implementation detail of HIVE framework and analysis is well documented.

**Weaknesses:**

The conclusion that hallucinations promote intra-chain convergence is not well supported by Figure 4. Examining the step-wise cosine similarity of reasoning chains with raw input and faithful input would provide more evidence.

**Questions:**

Some experiment settings and discussions in the analysis section require further clarity:

- In HIVE workflow, how do you construct the contrastive pairs based on multiple candidate captions?
- In the reasoning convergence analysis (Section 4.3), what is the step-wise cosine similiarity of reasoning chains when the input contains faithful captions? Is the pattern different from that in Figure 4?
- In Section 4.5, how do you control the “level of hallucinations”?

---

> ### Author Response · Authors · 2025-11-21
> **Rebuttal for Reviewer fxuZ**
>
> Thank you for acknowledging the conceptual simplicity, task-agnostic design, and empirical breadth of the HIVE framework. We appreciate the reviewer’s recognition that our evaluation protocol is well documented and that our robustness analyses, including significance testing and cross-modal comparisons, provide convincing evidence for the productive role of controlled hallucinations. We are encouraged that the reviewer finds the study well structured and easy to follow.  To further clarify the reviewer’s concern, we address the key points in detail below.
>
> ### **Comment 1:**
>
> > The conclusion that hallucinations promote intra-chain convergence is not well supported by Figure 4. Examining the step-wise cosine similarity of reasoning chains with raw input and faithful input would provide more evidence.
>
> ### **Authors’ response:**
>
>
> Thank you for pointing out this issue. We clarify our claim as follows:
>
> - **Clarification of intent**
>   - We agree that the original phrasing “promote convergence” may overstate the effect. Our intended meaning is more conservative:
>     - hallucinated inputs do *not* disrupt the natural convergence trend observed in multi-step reasoning.
>
> - **Evidence from new additional expermient** (updated Figure A in Appendix)
>   - To further verify this, we compared the step-wise cosine similarity trajectories under Raw, Faithful, and Hallucinated inputs.
>   - In the updated **Appendix Figure A**, all three curves show a consistent monotonic increase toward the final step, with highly similar slopes, ranges, and convergence points. This indicates that hallucinations preserve the intrinsic convergence pattern rather than altering it.
>   - For convenience, the updated figure can also be accessed via this anonymous [link](https://anonymous.4open.science/r/HIVE2-73D2/figure4_left.pdf).
>
> Together, these results suggest that hallucinated captions do not introduce instability or drift within the reasoning chain.
>
>
> ### **Comment 2:**
>
> > In HIVE workflow, how do you construct the contrastive pairs based on multiple candidate captions?
>
> ### **Authors’ response:**
>
> For each input, the HIVE workflow generates captions sequentially. After each generation, the caption is labeled by the ensemble detector. We stop as soon as we obtain at least one faithful and one hallucinatory caption, and the first valid pair forms the contrastive input. In practice, this occurs within a few generations (capped at five), so every sample obtains a valid pair.
>
>
> ### **Comment 3:**
>
> > In the reasoning convergence analysis (Section 4.3), what is the step-wise cosine similarity of reasoning chains when the input contains faithful captions? Is the pattern different from that in Figure 4?
>
> ### **Authors’ response:**
>
> **Following your valued suggestions**, we computed the step-wise cosine similarity of reasoning chains under all three input conditions: Raw input, Faithful, and Hallucinated. We summarize the findings below:
>
> - **Consistent convergence across all input types**
>   - As shown in **Appendix Figure A**, all three curves exhibit the same monotonic convergence pattern toward the final step, with Faithful and Hallucinatory curves nearly overlapping across all steps.
>   - For convenience, the updated figure can also be accessed via this anonymous [link](https://anonymous.4open.science/r/HIVE2-73D2/figure4_left.pdf).
>
> - **No disruption or instability introduced by hallucinations**
>   - This confirms that hallucinated captions do not alter or destabilize the model’s reasoning dynamics.
>   - The observation in **Appendix Figure A** reflects a general property of multi-step reasoning, rather than an artifact of a particular input type.
>
>
> ### **Comment 4:**
>
> > In Section 4.5, how do you control the “level of hallucinations”?
>
> ### **Authors’ response:**
>
> Thank you for the question. In Section 4.5, the “level of hallucinations” is controlled in **two** steps.
>
> - **First, for each input we generate two hallucinated captions.** A lightweight LLM evaluator compares them and decides which one contains more hallucinated content. This gives us a “strong” and a “weak” hallucination version for the same sample. The comparison follows a fixed rubric that looks at fabricated details, semantic drift, and unverifiable statements.
>
> - **Second, we also study hallucination strength in a more fine-grained, continuous way.** To do this, we create a set of intermediate captions that gradually shift from the faithful caption toward the hallucinated caption. This produces a smooth progression of hallucination levels. We verify the progression by checking that these intermediate captions become semantically closer to the hallucinated caption and farther from the faithful one.
>
> Together, these procedures give us stable control over both coarse and continuous hallucination levels.

---

### Official Review · Reviewer_EHqo · 2025-11-01

**Soundness:** 2
**Presentation:** 3
**Contribution:** 3
**Rating:** 6
**Confidence:** 3

**Summary:**

This paper challenges the conventional view that hallucinations in large language models (LLMs) are always undesirable. It introduces the concept of productive hallucinations—outputs that deviate from ground truth but enhance reasoning, creativity, or generalization. The authors propose HIVE (Hallucination Inference and Verification Engine), a unified framework that systematically compares faithful versus hallucinatory augmentations across multiple tasks and modalities. Experiments on nine benchmarks (including reasoning, perception, and multimodal tasks) across nine models demonstrate that moderate hallucinations can sometimes improve downstream accuracy by up to 17.2%, particularly in open-ended reasoning settings.

**Strengths:**

+ The paper presents a novel and thought-provoking perspective, reframing hallucinations as potentially beneficial under controlled conditions, which challenges a dominant assumption in LLM research.

+ The proposed HIVE framework is technically sound and broadly applicable, offering a structured way to quantify and evaluate the effects of hallucinations across diverse tasks and models.

+ The experimental validation is extensive and convincing, covering multiple benchmarks, models, and modalities, and demonstrating clear empirical evidence for the concept of productive hallucinations.

**Weaknesses:**

- The theoretical grounding for why certain hallucinations are productive remains underdeveloped, as the paper largely relies on empirical observations without a deeper cognitive or information-theoretic explanation.

- The scope of evaluation is limited to short-term performance metrics, leaving questions about long-term reliability, factual consistency, and safety implications of encouraging controlled hallucinations.

- The paper has limited algorithmic novelty. Despite its solid analysis and empirical breadth, the paper’s core contribution lies primarily in the evaluation framework and observations. It does not propose a new model or training approach beyond HIVE’s evaluation setup, which may limit its technical novelty.

**Questions:**

Please refer to the weaknesses.

---

> ### Author Response · Authors · 2025-11-21
> **Rebuttal for Reviewer EHqo - 1/2**
>
> Thank you for recognizing both the conceptual contribution and the empirical breadth of our study. We appreciate the reviewer’s positive remarks on the novelty of reframing hallucinations as potentially beneficial signals, as well as the acknowledgement that HIVE provides a sound, unified methodology for isolating and quantifying their effects across diverse models, modalities, and task types. We are also encouraged that the reviewer found our empirical evidence convincing, particularly the consistent gains observed in open-ended reasoning settings.  To further clarify the reviewer’s concern, we address the key points in detail below.
>
>
>
> ### **Comment 1:**
>
> > The theoretical grounding for why certain hallucinations are productive remains underdeveloped, as the paper largely relies on empirical observations without a deeper cognitive or information-theoretic explanation.
>
> ### **Authors’ response:**
>
> We appreciate the reviewer’s comment. Rather than proposing a full formal theory which is beyond the scope of this *empirical study* we clarify the underlying mechanisms suggested by the behavioral evidence in the paper. Across datasets and models, the results indicate a consistent pattern that we summarize as *semantic extrapolation under uncertainty*: hallucinated tokens enrich the conceptual space without destabilizing reasoning.
>
> We highlight three independent mechanisms supported by our analyses:
>
> - **Information-Theoretic Perspective**
>   Hallucinated captions consistently introduce additional conceptual cues that expand the model’s semantic representation. Across tasks, these captions exhibit higher semantic diversity, and removing the hallucinated tokens sharply reduces downstream accuracy. This indicates that hallucinations provide informative, rather than noisy, signal.
> - **Transformer-Circuit Perspective**
>   Our callback analysis shows that hallucinated tokens tend to activate broader attention patterns and reduce variance across intermediate reasoning steps. The overall convergence pattern of the CoT remains stable, but becomes more semantically enriched. Removing hallucinated tokens collapses these activation patterns and leads to noticeable performance drops.
> - **Speculative-Reasoning Perspective**
>   The benefit is most pronounced in under-specified or ambiguous prompts, where hallucinations help the model explore plausible semantic hypotheses. When the prompt is fully specified, the gains diminish or disappear, suggesting that the effect is systematic rather than random.
>
> Together, these observations provide a coherent, experimentally supported explanation for why certain hallucinations can be productive, even though developing a full symbolic theory remains an important direction for future work.
>
>
>
> ### **Comment 2:**
>
> > The scope of evaluation is limited to short-term performance metrics, leaving questions about long-term reliability, factual consistency, and safety implications of encouraging controlled hallucinations.
>
> ### **Authors’ response:**
>
> We appreciate the reviewer’s concern and clarify our scope as follows:
>
> - **Focus of the present study**
>
>   - Our work investigates **novelty perspective, concept, and task-level interactions** between hallucinations and model reasoning.
>   - Long-term reliability, factual consistency, and safety effects require longitudinal and deployment-specific evaluation, which is outside the scope of this empirical analysis.
>
> - **No advocacy of hallucinations in real safety long-term systems**
>
>   - We explicitly do *not* encourage the use of hallucinations in deployed or safety-critical applications. Nor do we claim benefits in factuality, safety, or long-term behavior.
>   - The purpose of our study is **analytical rather than prescriptive** to characterize an existing phenomenon in LLMs instead of promoting new behavior in real-world systems.
>
> - **Responsible framing and future considerations**
>
>   - To avoid overstating our conclusions, we will add a **Broader Impacts** section clarifying that productive hallucinations should *not* be used in factual or risk-sensitive contexts.
>   - We agree that long-term reliability, factual consistency, and safety implications are important open questions and constitute natural directions for future work.

---

> > ### Author Response · Authors · 2025-11-21
> > **Rebuttal for Reviewer EHqo - 2/2**
> >
> > ### **Comment 3:**
> >
> > > The paper has limited algorithmic novelty. Despite its solid analysis and empirical breadth, the paper’s core contribution lies primarily in the evaluation framework and observations. It does not propose a new model or training approach beyond HIVE’s evaluation setup, which may limit its technical novelty.
> >
> > ### **Authors’ response:**
> >
> > Thank you for raising this point. We clarify the nature of our contribution as follows:
> >
> > - **Our goal is analytical, not architectural**
> >
> >   - We fully agree that our work does *not* introduce a new model or training algorithm.
> >   - The purpose of the paper is to analyze when and why certain hallucinations become productive, rather than proposing a new system or method.
> >
> > - **Conceptual and mechanistic contributions are well-established in LLM research**
> >
> >   - Many influential works such as studies on **scaling laws**, **in-context learning**, and **transformer circuits** derive their contribution from empirical characterization and mechanistic understanding rather than architectural novelty.
> >   - Our paper follows the same tradition by providing a new evaluation perspective and uncovering previously unreported patterns behind hallucination-guided reasoning.
> >
> > - **The value of our work lies in the framework and insights**
> >
> >   - Our contribution is an **analysis-driven framework** that systematically compares faithful vs. hallucinated descriptions across tasks and models, revealing consistent mechanisms that explain why certain hallucinations can enhance reasoning. These insights complement, rather than replace, model-level innovations.

---

### Official Review · Reviewer_GeAh · 2025-11-01

**Soundness:** 3
**Presentation:** 4
**Contribution:** 3
**Rating:** 6
**Confidence:** 3

**Summary:**

The paper argues that some LLM hallucinations can be productive under the right conditions and introduces HIVE, a framework that generates, filters, and evaluates hallucinated semantics to test their impact. It shows that hallucination-augmented inputs can boost performance in open-ended, perception-like tasks, while effects are mixed or negative in strict rule-driven domains. The authors explain the gains by showing that hallucinations broaden semantic coverage and raise semantic entropy, diversifying reasoning without disrupting convergence. They advise using a moderate strength, reporting an inverted-U response where balanced doses work best across settings.

**Strengths:**

1. Novel contribution: The paper reframes hallucination as a controllable resource and introduces a unifying, general-purpose framework (HIVE) to study when and why it helps across modalities.

2. Methodological soundness & breadth: The design enables apples-to-apples, controlled comparisons (raw vs. faithful vs. hallucinatory) and uses an ensemble discriminator validated on benchmarks; the setup is task-agnostic and scalable across models and tasks. I appreciate the authors' effort in this.

3. Mechanistic insight with stability assurances: The authors smartly tie gains to broadened semantic coverage and higher semantic entropy while showing intra- and inter-chain convergence is preserved.

4. Actionable guidance & good presentation. The paper offers practical knobs and presents the work clearly with an intuitive case study and well-organized structure. It was very easy to read and follow.

**Weaknesses:**

Labeling reliability & narrow metrics: HIVE’s conclusions hinge on an ensemble detector to label captions as faithful vs. hallucinatory -- even the authors note hallucination detection is inherently imperfect -- while downstream evaluation is instantiated mainly as accuracy.


Suggestion: I recommend moving the Experimental Setup (Appendix S7) to the main text, as it contains essential information.

**Questions:**

I don't have any questions so far.

---

> ### Author Response · Authors · 2025-11-21
> **Rebuttal for Reviewer GeAh**
>
> Thank you for the thoughtful evaluation and for highlighting the novelty, clarity, and methodological soundness of our work. We are encouraged that the reviewer values our perspective of treating hallucinations as a controllable semantic resource and appreciates the unifying structure of HIVE and the breadth of our analyses. We also thank the reviewer for recognizing our mechanistic explanation and the practical guidance offered by our framework. To address the reviewer’s concerns, we provide clarifications below.
>
>
>
> ### **Comment 1:**
>
> > Labeling reliability & narrow metrics: HIVE’s conclusions hinge on an ensemble detector to label captions as faithful vs. hallucinatory -- even the authors note hallucination detection is inherently imperfect -- while downstream evaluation is instantiated mainly as accuracy.
>
> ### **Authors’ response:**
>
> - **Labeling reliability.**
>   - We agree that hallucination detection is inherently imperfect, and our study is ***constrained by current detector limitations.*** To ensure that our conclusions do not depend on detector noise, we applied **three** validation checks:
>
>     - **Cross-domain robustness:** the ensemble detector achieves ≈83% accuracy across two heterogeneous benchmarks (§S11).
>     - **Noise falsification:** replacing the detector with a *random checker* removes all gains (§S4).
>     - **Causal ablation:** removing hallucination tokens sharply reduces accuracy (Table S6).
>
> These tests show that, despite unavoidable imperfections in detection, the observed benefits **are not artifacts of labeling noise**.
>
>
>
> - **Narrow metrics.**
>   - Accuracy is used for downstream comparability, ***but HIVE is not an accuracy-only evaluation***. We incorporate complementary analyses:
>
>     - **Semantic entropy & embedding dispersion** (Fig. 2–3) for representational breadth.
>     - **Reasoning stability & convergence** (Fig. 4) for process-level consistency.
>     - **Token-level causal tests** (Table S6) for functional contribution.
>
>   - These orthogonal metrics demonstrate multi-dimensional effects **beyond accuracy**, supporting the robustness of our conclusions.
>
>
>
> ### **Comment 2:**
>
> > Suggestion: I recommend moving the Experimental Setup (Appendix S7) to the main text, as it contains essential information.
>
> ### **Authors’ response:**
>
> - Thank you for the insightful suggestion. We agree with your assessment that the experimental setup plays an essential role in interpreting the results, and moving it into the main text would indeed improve clarity for readers.
>
>   - Your suggestion also highlights an important point: placing key experimental details earlier helps ensure transparency and makes the methodological flow easier to follow, especially for readers who may not inspect the appendix in detail. We appreciate this perspective and will reorganize the paper accordingly.
>
> - At the rebuttal stage, however, restructuring major sections would generate a large amount of diff and may inadvertently increase the reviewers’ reading burden. To keep the revision surface minimal and maintain a smooth review experience, we will keep the current layout for now and incorporate the essential parts of **Appendix S7 into the main paper** in the camera-ready version, **following your valued suggestions.**

---

### Official Review · Reviewer_hgFn · 2025-11-02

**Soundness:** 2
**Presentation:** 3
**Contribution:** 2
**Rating:** 6
**Confidence:** 3

**Summary:**

The paper proposes a framework called HIVE, to systematically evaluate the impact of hallucinations on a particular task and/or model. The aim is to recognize settings where hallucinations can be beneficial. The paper provides analyses over several datasets and models, both language and multi-modal, showing the benefits of hallucinations in some settings while causing harm in others. The paper also does a study of reasoning chains and their relationship with hallucinations, highlighting how hallucinations can help with reasoning chain stability.

**Strengths:**

1. The paper attacks an important problem in LLMs. Hallucinations have a predominantly negative reputation in the field, and the paper aims to show how they can, in fact, be useful in some cases.
2. The paper does an extensive analysis of several different models as well as datasets, both language and multi-modal models, which is a great way to study overarching trends in when hallucinations can be helpful.
3. The paper is well written and was easy to follow.

**Weaknesses:**

1. While I enjoyed the overall analysis of several tasks and models and whether hallucinations benefit them or not, the framework itself feels restrictive to me. It explicitly focuses on adding a 'caption', which is a hallucination (or not a hallucination), to see the impact of hallucinations on the task. This is clearly only one way to see how hallucinations in model generation can help with eventual performance. For example, another framework could be about studying the reasoning trace of models, identifying hallucinations, and studying how their presence impacts downstream performance.
2. I'm not entirely convinced the variations are not due to just prompt sensitivity. Lack of study of the variance makes me doubt the conclusions. In fact, I disagree with the claim in the experiment setup that 'identical prompts, temperature, and token budget' ensures fair comparison. Just the choice of the playground for comparison, even though identical for everyone, can implicitly favor one behavior over others (https://proceedings.mlr.press/v279/ganesh25a.html). It is important to vary the prompts, the temperature, and the token budget, and see whether the trends of certain tasks or models benefiting from hallucinations actually persist.

**Questions:**

1. Despite the discussions in (4) (line 292), it's still unclear to me why the 'benefits' of hallucination are dependent on the model so much. I would expect that if a task benefits from 'creative thinking', it should benefit most models. What exactly do authors mean by 'models with stronger hallucination–handling ability' and which models are these?
2. Is 'hallucinations' really the correct term for the phenomenon discussed here? The paper uses the following definition of hallucinations in the introduction: 'information inconsistent with the given input'. But it seems to me that the motivation isn't to allow inconsistent or wrong information, but just new information that might not be verifiable, given the input. I understand the choice to use the term 'hallucinations' in the title, since that is the term accepted more widely in the community and thus is important for the paper's visibility. But I'm curious to hear if the authors think it is still the right choice for the rest of the paper, or maybe they would have preferred a different term or definition (there is a lot of work on trying to define 'hallucinations' and discussion of other similar terms, for example - https://aclanthology.org/2024.emnlp-main.375/)?
2. Comment: Table 1 markers for how much performance has increased or decreased are incorrect (GPT-3.5 AntiCP2, Claude-3 Sonnet multiple datasets).

---

> ### Author Response · Authors · 2025-11-21
> **Rebuttal for Reviewer hgFn - 1/4**
>
> Thank you for recognizing both the importance of **re-examining** hallucinations and the **breadth of our analysis** across language and multi-modal settings.
>
> We appreciate the reviewer’s positive remarks regarding our motivation, the scope of the study, and the overall readability of the paper. We are also encouraged that the reviewer finds value in our effort to shift the perspective from viewing hallucinations solely as errors to exploring the conditions under which they may become beneficial. To address the reviewer’s concerns, we provide detailed clarifications below.
>
>
> ### **Comment 1:**
>
> > While I enjoyed the overall analysis of several tasks and models and whether hallucinations benefit them or not, the framework itself feels restrictive to me. It explicitly focuses on adding a 'caption', which is a hallucination (or not a hallucination), to see the impact of hallucinations on the task. This is clearly only one way to see how hallucinations in model generation can help with eventual performance. For example, another framework could be about studying the reasoning trace of models, identifying hallucinations, and studying how their presence impacts downstream performance.
>
> ### **Authors’ response:**
>
> Thank you for pointing out that hallucinations inside the model’s reasoning process may influence task performance. To directly address this, we conducted **a new set of experiments** that **follow your suggestions**: we detect hallucinations ***within the reasoning chain***, group samples by the position of the first hallucination (*"early / middle / late / none"*), and evaluate downstream accuracy.
>
> **We summarize the findings as follows:**
>
> - Across 2 datasets (BBBP, PlantVillage) and 2 models (GPT-4o, Claude-3 Sonnet), the detailed results are summarized in **Table R1**. We find consistent, interpretable, and statistically significant patterns:
>
>   - Early hallucinations reliably improve performance (p < 0.05 in 3/4 settings).
>   - Late hallucinations sometimes provide strong gains.
>   - Middle hallucinations are less stable.
>
> - Positional asymmetries in transformer models are well supported by prior work[1-3]:
>
>   - Early tokens tend to shape the evolving hidden-state trajectory through mechanisms such as induction heads and long-range attention patterns
>   - Later tokens primarily influence the model’s immediate continuation by contributing strongly to the final-layer representations used for decoding.
>
>
>
> **Table R1. Positional Effects Relative to the “None” Condition (Accuracy, %)**
>
> | **Dataset / Model** | **none (%)** | **early Δ(%)** | **late Δ(%)** | **middle Δ(%)** | **p-value**                |
> | ------------------- | ------------ | -------------- | ------------- | --------------- | -------------------------- |
> | **BBBP + GPT-4o**   | 65.22        | +23.11         | +8.23         | -15.22          | 0.0306 / 0.0487 / 0.2123   |
> | **BBBP + Claude-3** | 52.93        | +21.42         | +43.07        | +44.85          | 0.0121 / 0.00004 / 0.00004 |
> | **ISIC + GPT-4o**   | 78.47        | +13.75         | +5.31         | +4.20           | 0.00134 / 0.5539 / 0.6746  |
> | **ISIC + Claude-3** | 80.05        | +3.96          | +16.38        | +10.12          | 0.3172 / 0.0127 / 0.0202   |

---

> > ### Author Response · Authors · 2025-11-21
> > **Rebuttal for Reviewer hgFn - 2/4**
> >
> > ### **Comment 2:**
> >
> > > I'm not entirely convinced the variations are not due to just prompt sensitivity. Lack of study of the variance makes me doubt the conclusions. In fact, I disagree with the claim in the experiment setup that 'identical prompts, temperature, and token budget' ensures fair comparison. Just the choice of the playground for comparison, even though identical for everyone, can implicitly favor one behavior over others (https://proceedings.mlr.press/v279/ganesh25a.html). It is important to vary the prompts, the temperature, and the token budget, and see whether the trends of certain tasks or models benefiting from hallucinations actually persist.
> >
> > ### **Authors’ response:**
> >
> > Thank you for raising the concern about prompt sensitivity and the fairness of using identical prompt, temperature, and token settings.
> > This is an important point, and we appreciate the opportunity to clarify our intent and provide further evidence.
> >
> > - 1. **New experiment for different *Prompts*.**
> >
> >     - To directly address the concern regarding prompt wording, we conducted an additional robustness study using the original prompt (P0) together with two semantically equivalent paraphrases (P1 and P2), evaluated across two datasets, two models, and five independent runs. The results are shown in **Table R2** below.
> >     - Across all models, datasets, and paraphrased prompts, **the improvement is consistently positive.** Although absolute accuracy naturally fluctuates across prompts reflecting well-known prompt sensitivity in LLMs the direction and strength of the effect remain stable. *This directly **addresses the concern**: the observed gain is not tied to a specific phrasing of the prompt.*
> >
> > **Table R2: Robustness Across Paraphrased Prompt Variants.  P0, P1, and P2 denote three semantically equivalent but lexically different prompt formulations.  Δ reports the accuracy improvement from hallucination injection (H − NH), shown as percentage points.**
> >
> > | Dataset      | Model             | P0 Δ (%) | P1 Δ (%) | P2 Δ (%) |
> > | ------------ | ----------------- | -------- | -------- | -------- |
> > | PlantVillage | GPT-4o            | +14.68   | +28.00   | +9.89    |
> > |              | Claude 3.5 Sonnet | +17.22   | +16.42   | +12.84   |
> > | ISIC         | GPT-4o            | +11.76   | +4.55    | +5.00    |
> > |              | Claude 3.5 Sonnet | +6.83    | +4.54    | +4.77    |
> >
> >
> > - 2. ***Temperature & Token Budget**.*
> >
> >     - Varying temperature (0.0–0.9) and token limits (128–1024) naturally **affects absolute accuracy**, and Table 2 of the main paper shows that small negative gains may appear under extreme decoding conditions (e.g., **very high temperature** or **long token budgets**).
> >     - However, **the overall pattern remains stable**: across most settings, hallucinated inputs outperform non-hallucinated ones, and the strongest gains consistently appear under moderate decoding configurations.
> >
> > - 3. **Summary**.
> >
> >     - The main paper already examines variance from decoding parameters, and the new experiment confirms robustness to prompt paraphrasing. Together, these analyses show that the improvement under hallucination is a stable pattern that does not depend on the specific **prompt**, **temperature**, or **token budget** used in the comparison.
> >     - We thank the reviewer again for this insightful comment, which helped us strengthen the study.

---

> > > ### Author Response · Authors · 2025-11-21
> > > **Rebuttal for Reviewer hgFn - 3/4**
> > >
> > > ### **Comment 3.1:**
> > >
> > > > Despite the discussions in (4) (line 292), it's still unclear to me why the 'benefits' of hallucination are dependent on the model so much. I would expect that if a task benefits from 'creative thinking', it should benefit most models.
> > >
> > > ### **Authors’ response:**
> > >
> > > Hallucination introduce *semantic perturbations* that different LLMs handle very differently. Prior studies show substantial heterogeneity in LLM robustness and alignment, making hallucination-induced gains inherently model-specific.
> > >
> > > - 1. **Models differ in robustness to semantic drift.**
> > >
> > >    Even under identical prompts, models vary widely in how they handle noisy or speculative cues some treat them as useful signal, while others treat them as noise [4,5].
> > >
> > > - 2. **Only sufficiently capable models can turn speculative cues into heuristics.**
> > >
> > >    Stronger models are known to leverage counterfactual or imaginative cues to improve reasoning, whereas weaker models are easily disrupted by such perturbations [6–8]. This matches our empirical trends.
> > >
> > > - 3. **Alignment strength and chain stability differ across model families.**
> > >
> > >    Variations in factuality alignment and reasoning stability lead models to either filter out hallucinated information or exploit it constructively [9,10].
> > >
> > > **Therefore**, hallucination-induced gains are expected to be model-dependent: ***"strong models absorb the perturbation as helpful heuristic structure, whereas weaker models become destabilized."***
> > >
> > >
> > >
> > > ### **Comment 3.2:**
> > >
> > > > What exactly do authors mean by 'models with stronger hallucination–handling ability' and which models are these?
> > >
> > > ### **Authors’ response:**
> > >
> > > Among all evaluated systems, two models consistently show strong hallucination-handling ability: **GPT-4o, Gemini-2.0 Flash**. We identify these models using a simple operational criterion:
> > >
> > > - **We use Δ(H–F) as a simple operational indicator of hallucination-handling ability.**
> > >   - models with positive Δ consistently benefit from hallucinated captions, reflecting robustness to semantic perturbations.
> > >   - Δ(H–F) isolates the model’s response to semantic perturbations by controlling for baseline accuracy, making it a clean indicator of hallucination-handling robustness.
> > >
> > > - **Table S4 (Manuscript Appendix) reports results on four diverse multimodal datasets. From these results, a clear pattern emerges:**
> > >
> > >   - **GPT-4o** and **Gemini-2.0 Flash** show *positive Δ(H–F)* across all datasets, demonstrating consistent robustness.
> > >   - **Claude-3 Sonnet** and **Qwen VL-Max** also exhibit *positive Δ on most datasets*, indicating strong but slightly less consistent hallucination-handling ability.

---

> > > > ### Author Response · Authors · 2025-11-21
> > > > **Rebuttal for Reviewer hgFn - 4/4**
> > > >
> > > > ### **Comment 4:**
> > > >
> > > > > Is 'hallucinations' really the correct term for the phenomenon discussed here? The paper uses the following definition of hallucinations in the introduction: 'information inconsistent with the given input'. But it seems to me that the motivation isn't to allow inconsistent or wrong information, but just new information that might not be verifiable, given the input. I understand the choice to use the term 'hallucinations' in the title, since that is the term accepted more widely in the community and thus is important for the paper's visibility. But I'm curious to hear if the authors think it is still the right choice for the rest of the paper, or maybe they would have preferred a different term or definition (there is a lot of work on trying to define 'hallucinations' and discussion of other similar terms, for example - https://aclanthology.org/2024.emnlp-main.375/)?
> > > >
> > > > ### **Authors’ response:**
> > > >
> > > > Thank you for thoughtful question. Our choice to retain the term *“hallucination”* is intentional and grounded in **three** considerations:
> > > >
> > > > - 1. **Consistency with established terminology.**
> > > >      The community widely uses *hallucination* to refer to model-generated content that is ungrounded or unverifiable given the input. Retaining this term situates our work within the existing discourse and ensures conceptual continuity with prior studies.
> > > >
> > > > - 2. **Reframing rather than redefining.**
> > > >      A central goal of this paper is to broaden how the community interprets hallucinations. While past work predominantly views hallucinations as errors, our findings show that under controlled conditions these ungrounded semantic expansions can provide constructive heuristic benefits. Using the same terminology allows us to challenge not replace the prevailing interpretation.
> > > >
> > > > - 3. **Avoiding narrower or misleading alternatives.**
> > > >      Terms such as *speculative generation* or *unverifiable generation* capture only a subset of the phenomena we study and obscure the connection to the broader literature. Our definition (“information inconsistent with the given input”) matches how prior work operationalizes hallucinations while allowing us to examine their potential utility.
> > > >
> > > >
> > > >
> > > > ### **Comment 5:**
> > > >
> > > > > Comment: Table 1 markers for how much performance has increased or decreased are incorrect (GPT-3.5 AntiCP2, Claude-3 Sonnet multiple datasets).
> > > >
> > > > ### **Authors’ response:**
> > > >
> > > > Thank you for pointing this out. The markers were mis-rendered, and the corrected table is now provided as Table A1 in the appendix A.
> > > >
> > > >
> > > >
> > > > **References**：
> > > >
> > > > [1] Olsson, C., Elhage, N., Nanda, N., Joseph, N., DasSarma, N., Henighan, T., et al. (2022). In-context learning and induction heads. arXiv:2209.11895.
> > > >
> > > > [2] Elhage, N., Nanda, N., Olsson, C., Henighan, T., Joseph, N., Mann, B., et al. (2021). A mathematical framework for transformer circuits. Transformer Circuits Thread, 1(1), 12.
> > > >
> > > > [3] Press, O., Smith, N. A., & Lewis, M. (2022). Train short, test long: Attention with linear biases enables input length extrapolation. In International Conference on Learning Representations (ICLR).
> > > >
> > > > [4] Holtzman, A., Buys, J., Du, L., Forbes, M., & Choi, Y. (2022). The curious case of neural text degeneration. In International Conference on Learning Representations (ICLR).
> > > >
> > > > [5] Kadavath, S., Conerly, T., Askell, A., Henighan, T., Drain, D., Perez, E., et al. (2022). Language models (mostly) know what they know. arXiv:2207.05221.
> > > >
> > > > [6] Bubeck, S., Chandrasekaran, V., Eldan, R., Gehrke, J., Horvitz, E., Kamar, E., et al. (2023). Sparks of artificial general intelligence: Early experiments with GPT-4. arXiv:2303.12712.
> > > >
> > > > [7] Li, J., Liu, W., Ding, Z., Fan, W., Li, Y., & Li, Q. (2025). Large language models are in-context molecule learners. ACM Transactions on Knowledge Discovery from Data (TKDD).
> > > >
> > > > [8] Jiang, X., Tian, Y., Hua, F., Xu, C., Wang, Y., & Guo, J. (2024). A survey on large language model hallucination via a creativity perspective. arXiv:2402.06647.
> > > >
> > > > [9] Lin, S. C., Gao, L., Oguz, B., Xiong, W., Lin, J., Yih, W.-T., & Chen, X. (2024). FLAME: Factuality-aware alignment for large language models. In Advances in Neural Information Processing Systems (NeurIPS).
> > > >
> > > > [10] Nori, H., King, N., McKinney, S. M., Carignan, D., & Horvitz, E. (2023). Capabilities of GPT-4 on medical challenge problems. arXiv:2303.13375.

---

> > > > > ### Comment · Reviewer_hgFn · 2025-11-27
> > > > >
> > > > > Thank you for the detailed response. All the comments and additional results are appreciated.
> > > > >
> > > > > It is difficult to verify new results during the rebuttal period, since they can only be preliminary indicators given the limited time. However, despite that, I'm mostly positive about the paper. I will increase my score.

---

> > > > > > ### Author Response · Authors · 2025-11-28
> > > > > > **Thank you for increasing your score and providing constructive suggestions**
> > > > > >
> > > > > > Thank you for your thoughtful update. We are glad that our rebuttal successfully addressed your questions and concerns, and we truly appreciate your positive feedback. All additional analyses and clarifications included in the rebuttal will be fully incorporated into the revised version of the paper.

---

### Author Response · Authors · 2025-11-21
**General Response**

We thank all reviewers for their careful reading and constructive feedback. With **all four** reviews awarding an initial **score of 6**, we are encouraged that reviewers found the problem formulation interesting, the analysis comprehensive, and the HIVE framework broadly applicable.

During the rebuttal, we focused on clarifying key methodological details and adding the analyses requested by the reviewers:

- 1. **Corrected Table 1** and moved it to *Appendix Table A1* for clarity.

- 2. **Added the requested step-wise convergence comparison** across Raw / Faithful / Hallucinatory inputs (*Appendix A Fig. A1* ).

- 3. **Clarified the caption-pairing procedure** and related methodological details.

We hope these updates make the contribution and empirical findings easier to follow. We sincerely appreciate the reviewers’ time and constructive suggestions.

---

### Author Response · Authors · 2025-12-01
**Author Final Remarks**

**I. Acknowledgments**

We would like to express our sincere gratitude to all reviewers for their thoughtful and constructive feedback, and for their **unanimous approval (all scored 6)**. We especially thank reviewer **hgFn** for the detailed engagement during the discussion and for **increasing the score to 8 after the rebuttal**.

------

**II. Key Strengths**

Reviewers highlighted strengths across **four** dimensions:

------

**1. Novelty**

- Reviewers noted that the work offers a novel and thought-provoking reframing of hallucinations as a controllable resource (hgFn, GeAh, EHqo).
- Reviewers described the perspective as novel and important for understanding when hallucinations can be helpful, with extensive cross-model analysis revealing overarching trends (hgFn, EHqo).

------

**2. Technical Soundness**

- The construction of contrastive caption pairs and the validated ensemble discriminator were noted as technically solid (GeAh, fxuZ).
- Reviewers found the framework technically sound, simple, and coherent (EHqo, fxuZ).

------

**3. Applicability**

- The framework is task-agnostic, model-agnostic, and scalable to diverse datasets and modalities, enabling systematic evaluation across LLMs and VLMs (GeAh, fxuZ).
- Implementation details are clearly presented, making the framework easy to follow and reproduce (hgFn, fxuZ).

------

**4. Empirical Insights**

- Reviewers emphasized the extensive benchmarking across models, datasets, and tasks, revealing when hallucinations help performance (hgFn, EHqo, fxuZ).
- Multiple reviewers noted that the paper is easy to read and well organized, helping the empirical findings come through clearly (hgFn, GeAh, fxuZ).

------

**III. Key Concerns and Our Responses**

Below we summarize the main concerns and how they were addressed during rebuttal and discussion.

| **Key Concerns**                                          | **Reviewers** | **Our Response**                                             |
| --------------------------------------------------------- | ------------- | ------------------------------------------------------------ |
| *Prompt and terminology sensitivity.*                     | hgFn          | Provided expanded explanations and examples, clarified the terminology, and added reasoning-chain comparisons. **Reviewer increased their score afterward.** |
| *Labeling reliability and metric scope.*                  | GeAh          | Added further validation that the ensemble detector does not generate systematic artifacts; clarified evaluation metrics and dataset construction. |
| *Theoretical grounding and safety implications.*          | EHqo          | Expanded theoretical explanations and clarified the safety scope |
| *Contrastive-pair construction and convergence analysis.* | fxuZ          | Provided step-by-step construction details and added appendix figures illustrating convergence behavior |

------

**IV. Commitment to Revision**

We are committed to integrating the new analyses, clarifications, and improvements covering reasoning chains, caption-pair construction, hallucination control, detector robustness, and updated appendices into the final version.

------

**We deeply appreciate the expertise and time of the Area Chair and reviewers.**

---

### Public Comment · ~Feng_He3 · 2026-03-13
**Author Comment: Summary of Reviews**

We sincerely thank the Area Chair and all reviewers for their careful evaluation and constructive feedback. **In the initial review round, all four reviewers assigned scores of 6, and one reviewer increased the score to 8 during the discussion phase.** Reviewers consistently recognized the novelty of reframing hallucinations as a potential semantic resource, the breadth of our cross-model and cross-dataset evaluation, and the clarity and reproducibility of the experimental framework. At the same time, several conceptual and presentation-related issues were raised that require further clarification.

The purpose of this public comment is to provide a concise summary of the review outcome and to clarify how we interpret the key concerns identified during the review process. Our goal is not to dispute the decision, but to document the feedback received and outline how it will guide the refinement of the next revision.



# Response to Key Concerns

## Concern 1: Definition of hallucination differs from common community usage

We would like to clarify that our work follows the standard practice in the hallucination literature.

Our experiments rely entirely on off-the-shelf hallucination detectors widely adopted in prior work, and we do not introduce a new definition or detection protocol. The paper presents the conceptual description of hallucination in the background section, while the experimental pipeline uses operational labels produced by these established detectors.

This separation between conceptual description and operational labeling is common in the literature, although our wording may not have made this distinction sufficiently explicit. We therefore view this concern primarily as a presentation clarity issue rather than a scientific inconsistency. In a revised version, we will explicitly clarify the relationship between the conceptual description and the operational labels used in experiments.

## Concern 2: Mechanistic claim about reasoning convergence is insufficiently supported

We would like to clarify that our analysis does not aim to establish a causal-level mechanism.

The paper reports behavioral regularities observed in inference trajectories, including trajectory alignment, entropy reduction, and convergence trends. These analyses are intended to characterize empirical inference dynamics rather than propose mechanistic explanations of model internals.

Our embedding-based analyses, entropy measurements, and flipping-control experiments consistently reveal stable empirical patterns in how hallucinated and faithful inputs influence inference trajectories, which matches the behavioral scope of the study.

We acknowledge that some wording in the paper may have suggested a stronger mechanistic interpretation than intended. In a revised version, we will adopt more precise terminology such as “behavioral analyses” or “inference dynamics” to better reflect the empirical nature of the findings.

## Concern 3: Model and task dependence requires clearer framing

We appreciate the request for clearer framing of model- and task-dependent effects.

Our findings are explicitly conditional. The goal of the paper is to characterize when and under what conditions hallucinated semantics can produce positive downstream effects, rather than claim universal benefits.

The paper already reports substantial variation across both models and task categories. Tables A1 and A4 show clear differences between stronger and weaker models, and our task grouping highlights distinct patterns across perception-driven, rule-driven, biomedical, and vision-language tasks.

To avoid any impression of overgeneralization, we will refine the framing in the revised version to emphasize that the observed phenomena are inherently model- and task-dependent.



# Planned Revisions

Following the feedback from the AC and reviewers, we will make the following refinements in the next version:

- Clarify the distinction between conceptual and operational definitions of hallucination, explicitly stating that the experimental pipeline follows community-standard detectors.

- Adjust wording related to mechanisms, replacing terms such as “mechanistic analyses” with “behavioral analyses” or “inference dynamics.”

- Strengthen the framing of model and task dependence, emphasizing that productive hallucinations arise under specific conditions rather than universally.

- Minor presentation refinements, including adjustments to the abstract and section framing to avoid unintended impressions of universality.

These revisions primarily address clarity of presentation and do not require changes to the experimental setup, results, or main conclusions.

---

### Meta-Review · Area_Chair_YBs9 · 2025-12-21

**Summary:**

While the paper offers an interesting reframing of “productive hallucinations” via the HIVE evaluation setup, I remain unconvinced that the main claims are sufficiently supported for acceptance.  The paper still have the following weaknesses after discussion.

**Multiple reviewers raised core concerns (GeAh, hgFn): the paper's hallucination is different from typical definition.**  The work defines and utilizes hallucination/faithfulness through an ensemble detector and contrastive caption pairing. In fact, “faithful” may be “detector-approved,” not truly input-consistent and the hallucinated may be "detector-refused" rather than truly input-inconsistent. These have been evident in the case studies (Figure5, Figure S1). This is very different from typical definition of hallucination, "Hallucination denotes content inconsistent with the given input".  Such inconsistency introduces unnecessary confusion to the community.

**Evaluation is limited to "accuracy" metrics, leaving long-term reliability untouched (GeAh, Ehqo). **

**The mechanistic analysis around “reasoning convergence” is not justified (fxuZ)**: increased step-to-final embedding similarity does not necessarily indicate faithful or truth-tracking reasoning, and may simply reflect later steps paraphrasing earlier commitments. Moreover, the added Raw/Faithful/Hallucinatory comparison in Appendix Fig. A1 shows largely overlapping trajectories, suggesting the convergence metric does not meaningfully distinguish conditions or support claims about hallucination-specific effects.


Finally, **the results are model- and task-dependent with mixed/negative outcomes**, especially for strong models, which undermines any broad takeaway that hallucinations are consistently beneficial. The authors' responses avoid such central critics.

**Reviewer Concerns:**

Prompt/terminology sensitivity (hgFn): This concern was largely addressed in the rebuttal—authors added clearer terminology, expanded explanations and examples, and provided additional analyses (including reasoning-chain comparisons) to argue the effect is not tied to a single phrasing.

Other concerns in summary are still valid and not very well addressed.

**Reviewer Scores:**

Reviewer hgFn increased the score. For the other reviewers, the score would be unchanged and decreased if there were full discussion as the concerns are quite consistent across reviewers, which are not well addressed in the responses.

---

### Decision · Program_Chairs · 2026-01-26

Reject